# A theory of learning data statistics in diffusion models, from easy to hard

**Lorenzo Bardone** [1]   **Claudia Merger** [2]   **Sebastian Goldt** [2]

## Abstract

While diffusion models have emerged as a powerful class of generative models, their learning dynamics remain poorly understood. We address this issue first by empirically showing that standard diffusion models trained on natural images exhibit a distributional simplicity bias, learning simple, pair-wise input statistics before specializing to higher-order correlations. We reproduce this behaviour in simple denoisers trained on a minimal data model, the mixed cumulant model, where we precisely control both pair-wise and higher-order correlations of the inputs. We identify a scalar invariant of the model that governs the sample complexity of learning pair-wise and higher-order correlations that we call the *diffusion information exponent*, in analogy to related invariants in different learning paradigms. Using this invariant, we prove that the denoiser learns simple, pair-wise statistics of the inputs at linear sample complexity, while more complex higher-order statistics, such as the fourth cumulant, require at least cubic sample complexity. We also prove that the sample complexity of learning the fourth cumulant is linear if pair-wise and higher-order statistics share a correlated latent structure. Our work describes a key mechanism for how diffusion models can learn distributions of increasing complexity.

## 1. Introduction

Introduced only ten years ago, diffusion models (Sohl-Dickstein et al., 2015; Ho et al., 2020; Song & Ermon, 2019) quickly reached state-of-the-art performance in generative modeling. Yet our theoretical understanding of why these models learn so efficiently remains limited compared to our

understanding of neural networks in "standard" supervised learning.

In supervised learning, a key result is that neural networks exhibit a *simplicity bias* – they first learn simpler features of their target before moving to more complex features. This effect can be seen both in time during training, or as a function of the training set size. Simplicity biases were shown first in the context of learning a target function over Gaussian inputs (Saad & Solla, 1995; Saxe et al., 2014; 2019; Abbe et al., 2023; Dandi et al., 2024; Berthier et al., 2025), in autoencoders (Kögler et al., 2024), and experimentally for image classification (Kalimeris et al., 2019); a similar effect can also be seen in the kernel regime (Farnia et al., 2018; Rahaman et al., 2019). More recently, a *distributional simplicity bias*, whereby neural networks first rely on pair-wise input statistics before exploiting higher-order correlations was shown both theoretically and experimentally in image classification (Ingrosso & Goldt, 2022; Merger et al., 2023; Refinetti et al., 2023; Bardone & Goldt, 2024) and next-token prediction (Rende et al., 2024; Belrose et al., 2024; Favero et al., 2025; Garnier-Brun et al., 2025).

Whether similar principles govern the *distributional* learning dynamics of diffusion models remains an open question. Here, we show through a combination of careful experiments and a rigorous analysis of SGD dynamics in simplified models that denoising diffusion models exhibit a distributional simplicity bias.

Our **main experimental contribution** is a demonstration of distributional simplicity bias in standard U-net based denoisers trained in a denoising diffusion paradigm on an image modeling task, see fig. 1 and section 2. Specifically, we find for the first $\approx 10^3$ steps of training, a U-Net trained to denoise CIFAR10 images achieves the same test loss on the CIFAR10 test set as it does on samples from a Gaussian distribution that have the same mean and covariance as the CIFAR10 images. In other words, the denoiser only relies on pair-wise correlations between pixels to denoise images up to that point in training. Only after about $10^3$ steps of SGD, the network starts to exploit the higher-order correlations between pixels, which is evidenced by the lower loss of the denoiser on real images, where such correlations are present, than on the Gaussian surrogate model. We discuss this experiment in more detail in section 2.

[1]Statistical Physics of Computation Laboratory, EPFL, Lausanne, Switzerland [2] Theoretical and Scientific Data Science Group, SISSA, Trieste, Italy. Correspondence to: Lorenzo Bardone <lorenzo.bardone@epfl.ch>.

*Proceedings of the 43$^{rd}$ International Conference on Machine Learning*, Seoul, South Korea. PMLR 306, 2026. Copyright 2026 by the author(s).

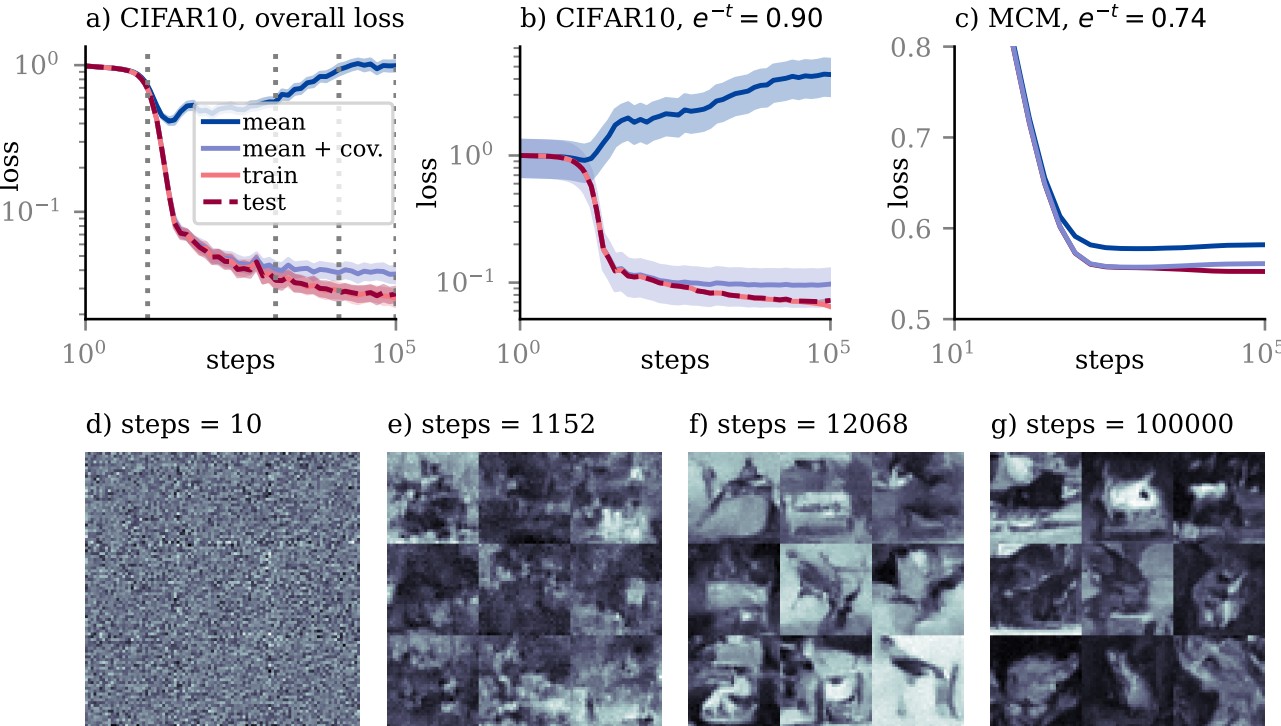

*Figure 1.* Sequential learning in diffusion models. a) Test loss of diffusion model and loss on CIFAR-10 clones during training. Vertical dotted lines mark training stages of images generated from the model shown in panels d)-g). All curves are averages over 3 initializations of the network models and $5 \cdot 10^3$ test data. Shaded areas report standard deviation over random initialization. Panel b) reports the same as a), but for denoising samples with fixed level of noise $x = e^{-t}x_0 + \sqrt{1 - e^{-2t}}z$ where $x_0$ is a data point and noise $z \sim \mathcal{N}(0, \text{Id})$. c) Test loss of a neural network trained on the mixed cumulant model with dimension $d = 10^2$ a fixed level of noise, evaluated on clones of the data set. All curves in panel c) are averages over 5 initializations of the network model and $10^4$ test data. d)-g) Samples generated from U-nets (Ronneberger et al., 2015) on CIFAR-10 at various training stages.

How can we describe this distributional simplicity bias theoretically? While several works have examined the dynamics of score denoising in simplified models (Li et al., 2024; Bonnaire et al., 2025; George et al., 2025; Merger & Goldt, 2025; Wang & Pehlevan, 2025), they either consider random feature denoisers, model data as drawn from a Gaussian distribution, or characterize an inductive bias toward Gaussian/linear denoisers during generalization (Li et al., 2024), and hence cannot account for genuine feature learning of higher-order input correlations. Feature learning in non-linear autoencoders, arguably the simplest class of denoiser neural networks, on a reconstruction task was analysed by Refinetti & Goldt (2022); Cui & Zdeborová (2023). More recently, Cui et al. (2024; 2025) extended this approach to analyse the learning dynamics of auto-encoders trained in a diffusion paradigm at linear sample complexity. Here, we instead characterise the learning dynamics in a solvable model of the learning dynamics of score diffusion by identifying a key invariant that governs the sample complexity at which a denoising diffusion model recovers information carried by pair-wise and higher-order correlations, going beyond linear sample complexity.

For our **main theoretical results**,

1. we prove nearly sharp thresholds for the number of samples required by a simple denoiser to learn from higher-order correlations (section 4.1);

2. we rigorously establish the distributional simplicity bias of denoising diffusion models by proving a separation of timescales between learning pair-wise and higher-order correlations (section 4.2);

3. we demonstrate that constraining SGD to the unit sphere, often considered a minor technical detail, is crucial for efficient learning in this model: unconstrained SGD can remain confined near the trivial solution, while spherical SGD exhibits successful learning dynamics, whose timescales we explicitly characterize (section 4.3).

4. we analyse this effect in more detail in the special case where we choose the optimal denoiser (section 4.4).

Our results thus describe a key mechanism for how denoising diffusion models learn distributions of increasing

complexity. On the technical level, we identify a scalar invariant of the loss that describes the initial stages of the learning dynamics, which we call the **diffusion information exponent** $k^*$ by analogy to similar invariants found for the single/multi-index problems like the information exponent (Ben Arous et al., 2021), which allows us to establish nearly sharp thresholds on the sample complexity of learning specific statistical features of the data.

## 2. A distributional simplicity bias in denoising diffusion models

Diffusion models are trained by adding increasing amounts of Gaussian i.i.d. noise to real data until the original data is no longer discernible from pure noise. A neural network is trained to reverse this process by predicting and removing the noise step by step. To learn to denoise data in this way, the model must make use of the statistics of the data it is trained on, i.e. it must infer the characteristic properties of the distribution of the data. After training, one starts from random noise and iteratively removes the noise predicted by the model to obtain new, realistic samples.

To measure which statistics of the data the model is exploiting, we evaluate the model on several clone datasets, which share statistical properties of the real dataset during training (Refinetti et al., 2023). Here, our original dataset are grayscale CIFAR-10 images (Krizhevsky, 2009), on which we train a U-net architecture (Ronneberger et al., 2015) using denoising score matching (Song & Ermon, 2019; Ho et al., 2020). Each clone is a data set of inputs sampled from a Gaussian, whose mean ("mean") or mean and covariance ("mean + cov") have been fitted to the images in CIFAR-10. We detail their generation and show examples of the clone data sets in section A.1.

The test loss measures the models' ability to predict the noise added to the images, both real images and those of the clones. If the model has equal performance on the real data and a clone dataset, this means that, despite the real data having a far richer statistic than the clone, the model has not learnt to use those statistics to predict the noise yet. In this case, we choose the clones to reproduce statistics that can be inferred from the data directly (the mean and the covariance).

Figure 1a) shows that as training progresses, the diffusion model specializes more and more: at first, the performance on all test sets (real data and clones) is equal. At later training stages, the models' performance on the more specialized clones improves, whereas its performance on less specialized clones stagnates, meaning that the model has learnt to exploit the statistics of the data that go beyond the one of the clones it outperforms. We show the samples obtained from these models at various training stages in fig. 1 d)-g).

We repeat the same experiment with the CelebA data set and find the same sequential learning behaviour, see section A.2. We report details on the training procedure in A.2.1. This experiment substantiates the claim that lower order statistics (i.e. mean and covariance) are learnt in the initial phases of learning, whereas higher-order statistics are learnt later. In the following, we will introduce a model which allows us to explain this sequential learning property of the neural network from lower to higher order statistics.

## 3. Setup for our theoretical analysis

We will analyse the dynamics of projected stochastic gradient descent (pSGD) for a simple, non-linear denoiser trained on inputs sampled from a non-Gaussian distribution. We now describe in detail the diffusion paradigm we analyse, the input distribution we use, and the denoiser we will train.

### 3.1. Denoising diffusion

We model the diffusion process following a standard approach, see for instance (Biroli & Mézard, 2023); all the details are in section B.3. The diffusion process is defined on a time interval $t \in [0, T]$, with $P_0$ the unknown distribution that we want to learn to sample from, and a distribution over latent variables $P_T \approx \mathcal{N}(0, \mathbb{1}_d)$. The dynamics are described by the following SDE:

$$\mathrm{d}x(t) = -x\mathrm{d}t + \mathrm{d}\mathcal{W}_t, \tag{1}$$

where $\mathcal{W}_t \in \mathbb{R}^d$ is a $d$ dimensional Wiener process. The solution at time $t$ can be written in distribution as:

$$x(t) \overset{\mathscr{D}}{=} e^{-t}x(0) + \sqrt{\Delta_t}z \tag{2}$$

where $x(0) \sim P_0$, $z \sim \mathcal{N}(0, \mathbb{1}_d)$ and $\Delta_t = 1 - e^{-2t}$. The goal in diffusion models is learning the score of the density at intermediate times, which is given by

$$\mathcal{F}_i(x, t) = \frac{\partial \log P_t(x)}{\partial x_i} \tag{3}$$

$$= -\frac{x_i - \mathbb{E}\left[x_i(0)|x(t) = x\right]e^{-t}}{\Delta_t}, \tag{4}$$

where the last equality, called *Tweedie's formula*, is at the core of the feasibility of diffusion models. It gives a recipe on how to approximate the score via empirical averages of the noised process.

The objective then becomes learning $\mathcal{F}$, which allows one to realize the reverse SDE (Anderson, 1982) and to generate new samples from pure noise. Here, we will not treat the generation process, but focus on learning the score from data. To do this, one usually uses a mean-square objective for a collection of fixed time intervals. Let us denote $\mathcal{S}_t^w(x)$ the approximated score that depends on weight $w$. The

loss function for diffusion time $t$ can be rewritten, up to a constant, as

$$\mathcal{L}(w) = \frac{1}{2} \mathop{\mathbb{E}}_{\substack{x_0 \sim \mathbb{P}_0 \\ z \sim \mathcal{N}(0, \mathbb{1}_d)}} \left\| S_t^w(x_0 e^{-t} + \sqrt{\Delta_t} z) + \frac{z}{\sqrt{\Delta_t}} \right\|^2,$$
(5)

see section B.3 for a derivation.

The loss in eq. (5) can be well approximated having just samples from $P_0$, which allows to estimate the integral over $x_0$. The additional Gaussian integral over $z$ in eq. (5) is a term that is peculiar to diffusion models. To perform the subsequent analysis, we must decompose the loss into Hermite polynomials. To this end, we use *Stein's lemma* (lemma B.4 in the appendix), following the approach of (Shah et al., 2023), to rewrite the loss and apply the Hermite decomposition.

## 3.2. Input distribution

We draw samples $x$ from the *mixed cumulant model* (MCM) of Bardone & Goldt (2024). The idea of the mixed cumulant model is to generate inputs that appear isotropically Gaussian in all directions except along the two vectors, or "spikes", $u, v \in \mathbb{R}^d$. This means that taking linear projections of inputs sampled from the mixed cumulant model along a fixed, random direction $w \in \mathbb{R}^d$, $\lambda^\mu = w \cdot x^\mu$, results in random variables $\lambda^\mu$ that follow a standard normal distribution with high probability. However, there are two special directions that a generative model needs to learn: first, the covariance spike $u$, along which inputs are still normally distributed, but with higher variance, defined by the signal-to-noise ratio $\beta_u$. The second special direction is the cumulant spike $v$: projecting inputs along this cumulant spike yields a non-Gaussian distribution; by constraining the variance of this distribution to be equal to one, we ensure that the cumulant spike $v$ can only be discovered using higher-order statistics of the inputs, making it harder to detect.

We construct samples $x^\mu$, $\mu \in [n]$ of the mixed cumulant model thus:

$$x^\mu = \sqrt{\beta_u} \lambda^\mu u + \sqrt{\beta_v} \nu^\mu v + z^\mu$$
(6)

where $\beta_u \in \mathbb{R}, \beta_v \in [0, 1]$ are constant *signal to noise ratios* that modulate the intensity of the signal, $\lambda^\mu \sim \mathcal{N}(0, 1)$ and $\nu^\mu \sim \text{Rademacher}(1/2)$ are latent variables, and $z^\mu \sim \mathcal{N}(0, \mathbb{1} - \beta_v v v^\top)$ is high-dimensional noise.

To connect with natural data distributions, one can think of $u$ as a dominant Fourier mode capturing low-frequency structure with roughly Gaussian projections, and of $v$ as a localized Gabor-like filter that captures the leading higher order components of natural image statistics (Hyvärinen et al., 2009).

## 3.3. Denoiser and learning algorithm

We consider the simplest architecture that is able to learn the score of a MCM with one non-Gaussian spike (the explicit derivation of the target score is in appendix B.4). The denoiser takes the form of a rank-1 non-linear autoencoder with a skip connection:

$$S_t^w(x) = -x - \sigma(w \cdot x) w,$$
(7)

where $w \in \mathbb{S}^{d-1}$ is the vector of trainable weights. Autoencoders of this type have been studied before for reconstruction (Refinetti & Goldt, 2022; Cui et al., 2025; Mendes et al., 2026) and for denoising diffusion (Cui et al., 2025).

To train the denoiser weight $w$, we update an initial weight $w_0 \sim \text{Unif}(\mathbb{S}^{d-1})$ via online projected SGD (pSGD):

$$\tilde{w}_{\tau+1} = w_\tau - \eta_d \nabla_{\text{sph}} \mathscr{L}(w_\tau, x_\tau), \quad w_{\tau+1} = \frac{\tilde{w}_{\tau+1}}{\|\tilde{w}_{\tau+1}\|}$$
(8)

where $\mathscr{L}(w, x)$ is the sample-wise loss that substitutes the average over $\mathbb{P}_0$ in eq. (5) with the evaluation on a sample $x \sim \mathbb{P}_0$. $\nabla_{\text{sph}}$ is the spherical gradient: $\nabla_{\text{sph}} f(w) = (\mathbb{1} - w w^\top) \nabla f(w)$. We analyse projected SGD rather than standard SGD mainly because it improves the network's ability to learn the score in our controlled setting. We discuss how using non-projected SGD impacts learning in section 4.3.

We simplify $\nabla_{\text{sph}} \mathscr{L}$, by expanding the square in eq. (5), differentiating with respect to $w$ and then applying Stein lemma B.4 to remove the integral over $z$. At each training time $\tau$, $\mathscr{L}$ is computed on a new independent sample $x_\tau \sim \mathbb{P}_0$. We obtain a formula that depends only on samples of $x \sim \mathbb{P}_t$, not $z$:

$$\nabla_{\text{sph}} \mathscr{L}_t(w, x) = (\mathbb{1}_d - w w^\top) x F_\sigma(x \cdot w)$$
(9)

where we have defined the *effective nonlinearity*

$$F_\sigma(x \cdot w) := \sigma''(x \cdot w) - \sigma'(x \cdot w) \sigma(x \cdot w) \\ - \sigma(x \cdot w) - \sigma'(x \cdot w) x \cdot w, \quad (10)$$

The detailed derivation is given in appendix section B.5.

## 4. Theoretical analysis of score denoising

Our goal is now to establish the sample complexities required to learn certain statistical structures of the inputs like the covariance and cumulant spikes $u$ and $v$ with the diffusion model. Our strategy will be to identify a scalar invariant that describes the initial stages of the learning dynamics, which we call the **diffusion information exponent** $k^*$ by analogy to similar invariants found for single/multi-index problems like the information exponent (Ben Arous et al., 2021), generative exponent (Damian et al., 2024), or the

leap index (Dandi et al., 2024). Here, we extend these ideas to the diffusion setting. The main technical difference compared to their setting is that they assumed a Gaussian distribution over inputs with identity covariance; here we analyse a non-isotropic, non-Gaussian input distribution, using the methodology of Bardone & Goldt (2024).

To highlight the difficulty of learning higher-order correlations using denoising diffusion, we start by considering a setting where inputs only carry non-trivial structure in their higher-order correlations, meaning that the covariance signal-to-noise ratio is $\beta_u = 0$. In this case, the only relevant order parameter to describe the learning dynamics is the overlap $\alpha = w \cdot v$ between the weight vector of the denoiser and the cumulant spike. This overlap is small at initialization, of order $\alpha = \Theta(1/\sqrt{d})$. We will say that the autoencoder has learnt the higher-order correlations if the autoencoder has "weakly recovered" the cumulant spike $v$, i.e. when the overlap $\alpha = w \cdot v \sim O(1)$. This transition from diminishing to macroscopic overlap $\alpha$ marks the exit of the search phase of stochastic gradient descent, and it often requires most of the runtime of online SGD (Ben Arous et al., 2021).

### 4.1. The diffusion information exponent determines the sample complexity for recovering data structure

We now introduce the *diffusion information exponent*, which determines the sample complexity required to learn correlations of different orders. We expand both the effective non-linearity $F_\sigma$ and the likelihood ratio

$$L_t := \frac{\mathrm{d}P_t}{\mathrm{d}\mathcal{N}(0, \mathbb{1}_d)}$$

in the Hermite basis (see definition B.2), with coefficients $(c_i^F)_{i \in \mathbb{N}}$ and $(c_j^L)_{j \in \mathbb{N}}$, respectively. The Hermite coefficients of $F_\sigma$ encode the properties of the loss function together with the chosen nonlinearity $\sigma$, while the Hermite coefficients of the likelihood ratio characterize the structure of the data distribution. The hardness of the inference task in the online regime is governed by how these two sequences of coefficients interact.

When the learning rate is sufficiently small, the noisy online dynamics are well approximated by the gradient flow of the population loss. Then the evolution of the overlap $\alpha_\tau$ is dominated by the leading non-vanishing contribution in the Hermite expansions, yielding

$$\alpha_{\tau+1} = \alpha_\tau + \eta_d \, c_{k^\star}^L \, c_{k^\star-1}^F \, \alpha_\tau^{k^\star-1} + O\left(\alpha_\tau^{k^\star}\right).$$

We define $k^\star$, the **diffusion information exponent**, as the smallest integer $k$ such that the $k$-th Hermite coefficient of the likelihood ratio and the $(k-1)$-th Hermite coefficient of $F_\sigma$ are both non-zero. Intuitively, $k^\star$ identifies the lowest-order statistical feature of the data that is both present in the

distribution and exploitable by the combination of the mean-squared error loss and the nonlinearity $\sigma$. As a consequence, starting from a random initialization, it takes on the order of $d^{k^\star-1}$ iterations—and hence samples, since we are in the online regime—to reach recovery of the spike $v$, as made precise in the propositions that follow.

*Assumption* 4.1 (Essential). $\sigma$ and $\mathbb{P}$ are such that $\nabla_{\mathrm{sph}}\mathcal{L}(\alpha)$ is strictly negative for all $\alpha \in (0, 1)$

*Assumption* 4.2 (Technical). Define

$$H_d(x, w) := \mathscr{L}(x, w) - \mathcal{L}(w)$$

We assume the following estimates hold for some $C_1 > 0$, $\varepsilon > 0$:

$$\sup_{w \in \mathbb{S}^{d-1}} \mathbb{E}\left[(\nabla_{\mathrm{sph}} H_d(x, w) \cdot v)^2\right] \leq C_1 \tag{11}$$

$$\sup_{w \in \mathbb{S}^{d-1}} \mathbb{E}\left[\|\nabla_{\mathrm{sph}} H_d(x, w)\|^{4+\varepsilon}\right] \leq C_1 d^{(4+\varepsilon)/2} \tag{12}$$

**Proposition 4.3** (Positive result). *Assume that $L_t(x \cdot v)$ is the likelihood ratio of a sub-Gaussian random variable, and $\sigma$ an activation function such that $F_\sigma$ satisfies assumption 4.1 and assumption 4.2. Denote with $k^*$ the information exponent of the loss $\mathcal{L}$ and let $\hat{n}(d, k^*)$ be a sample complexity threshold defined as:*

$$\begin{cases} \hat{n}(d, 1) = \omega(d) \\ \hat{n}(d, 2) = \omega(d \log^2 d) \\ \hat{n}(d, k) = \omega(d^{k-1} \log^2 d) & k \geq 3 \end{cases}$$

*then the application of $\hat{n}(d, k^*)$ steps of projected gradient descent with step size $\eta_d$ satisfying*

$$\frac{1}{\hat{n}} \ll \eta_d \ll \frac{1}{\sqrt{\hat{n}d}} \tag{13}$$

*starting from isotropic initialization $w \sim Unif(\mathbb{S}^{d-1})$ leads to:*

$$\lim_{d \to \infty} |v \cdot w(\hat{n}(d, k^*))| = 1. \tag{14}$$

*Where the limit holds in probability and in $L^p$ for all $p \geq 1$.*

Note that in the proposition we used the *little omega* notation, see appendix B.1

**Proposition 4.4** (Negative result). *In the setting of the previous propositions, if $n(d) = o(\hat{n}(d, k^*))$ and*

$$\eta_d = \begin{cases} O\left(\frac{1}{d}\right) & k^* = 1, 2 \\ \eta_d = O\left(\frac{1}{\sqrt{n(d)d}}\right) & k^* \geq 3 \end{cases}$$

*the online SGD with learning rate $\eta_d$ will fail to reach weak recovery:*

$$\lim_{d \to \infty} \sup_{\tau \leq n(d)} |v \cdot w(\tau)| = 0 \tag{15}$$

*where the limit is in probability and in $L^p$ for any $p \geq 1$.*

*Proof.* Propositions 4.3 and 4.4 are essentially corollaries of theorems 1.3 and 1.4 in (Ben Arous et al., 2021). We explain the details in appendix B.5. $\quad\square$

Together, propositions 4.3 and 4.4 show how the diffusion information exponent $k^\star$ governs the sample complexity of online SGD: the model can only recover the planted direction $v$ after it has seen a number of samples roughly on the order of $d^{k^*-1}$, and not before. For our setting given by eq. (6) with $\beta_u = 0, \beta_v > 0$, we find $k^* = 4$, meaning that recovery of the cumulant spike takes a number of samples larger than cubic in the dimension.

We note that in the setting of proposition 4.3 and 4.4, the lower bounds of Székely et al. (2024) for algorithmic detection of the cumulant apply, which suggest that the cumulant spike $v$ can be weakly recovered by a polynomial-time algorithm with $d^{k^*/2}$ samples, which is less than the sample complexity that we found for online SGD. In analogy to Gaussian single-index models, we expect that the smoothing techniques of Biroli et al. (2020); Damian et al. (2023) could reduce the sample complexity from $n \gtrsim d^{k^*-1}$ down to $n \gtrsim d^{k^*/2}$, at the cost of fine-tuning the activation function of the denoiser to the relevant cumulant of the data distribution; see Ricci et al. (2025) for an example of effect in the context of independent component analysis.

Even reducing the sample complexity to $d^{k^\star/2}$ would not fully account for the rapid separation from Gaussian clones observed in fig. 1. This gap likely reflects the highly idealized nature of our setting, where higher-order statistics are completely isolated; in the next section, we study how interactions between low and higher-order information qualitatively change this picture.

## 4.2. Simplicity bias in denoising diffusion

We now add back the covariance spike $\beta_u$ to our data model, which is carried by the covariance. This corresponds more closely to the setting found in real tasks, where both lower and higher order statistics are present in the data. The dynamics in this case are more complex, and the sample complexity required to learn the structure in the higher-order correlations depends on whether there are correlations among latent variables, as illustrated by the next proposition.

*Assumption 4.5.* The link function $\sigma$ is a thrice differentiable function, with bounded first, second and third order derivatives. Hence $F(z) = \sigma''(z) - \sigma'(z)\sigma(z) - \sigma(z) - \sigma'(z)z$ belongs to the space of square integrable functions with respect to the density of $\mathcal{N}(0, \mathbb{1})$ for all $t$ (being all sub-Gaussian distributions). Assume moreover that $\sigma$ is so that

$F$ satisfies the following conditions

$$c_1^F = \underset{z \sim \mathcal{N}(0,1)}{\mathbb{E}} [F(z)h_1(z)] > 0 \qquad (16)$$

$$c_3^F = \underset{z \sim \mathcal{N}(0,1)}{\mathbb{E}} [F(z)h_3(z)] < 0 \qquad (17)$$

**Proposition 4.6.** *Under assumption 4.5, consider pSGD eq.* (8) *dynamics trained on $\mathscr{L}$ defined as eq.* (9)*, with data distributed as a* mixed cumulant model *eq.* (6)*. Then:*

1. *with* independent latent variables $\lambda^\mu$, $\nu^\mu$ *and learning rate $\eta_d \to 0$ as $d \to \infty$, as long as $n = o_d\left(\min\left(\frac{d}{\eta_d^2}, d^3\right)\right)$, we have that $\lim_{d \to \infty} \sup_{\tau \le n} |w_\tau \cdot v| = 0$ in $L^p$ for every $p \ge 1$.*

2. *with a number of samples $n = \theta_d d$, with $\theta_d = \Omega(\log^2 d)$ and growing at most polynomially in $d$; step size $\eta_d$ chosen so that $\frac{1}{\theta_d} \ll \eta_d \ll \frac{1}{\sqrt{\theta_d}}$ pSGD reaches weak recovery in a time $\tau_u \le n$ i.e. there exists $\iota > 0$ independent of $d$ such that for $\tau \ge \tau_u$, with high probability $w_\tau \cdot u \ge \iota$. Moreover, in the case of positive correlation of latent variables $\mathbb{E}[\lambda^\mu \nu^\mu] > 0$, conditioning on having matching sign at initialization: $(v \cdot w_0)(u \cdot w_0) > 0$, weak recovery is achieved also for the cumulant spike $v$ in a time $\tau_v \le n$.*

The first part of proposition 4.6 is a negative result: the cumulant spike cannot be recovered at linear sample complexity. The second part of this statement is instead a positive result: pSGD weakly recovers the covariance spike $u$, and hence learns about the pair-wise statistics, in quasi-linear sample complexity. For the cumulant spike $v$ instead, we find that if the latent variables $\lambda^\mu$ and $\nu^\mu$ are uncorrelated, pSGD will need at least $d^3$ samples to weakly recover $v$ (as for single spike models, see proposition 4.3), and hence learn about higher-order correlations. This clear separation of timescales for the recovery of $u$ and $v$ rigorously establishes the distributional simplicity bias: the model learns pair-wise statistics (long) before higher-order correlations. However, if latent variables have a positive correlation, pSGD will recover the spike $v$ with $\Theta(d\,\mathrm{polylog}(d))$ samples. A similar speed-up due to correlated latent variables was found by Bardone & Goldt (2024) for supervised classification. We provide the proof in section B.6. We show an example of a neural network trained on the mixed cumulant model in fig. 1 c). This model exhibits the same sequential learning property as conventionally trained diffusion models on image data, see fig. 1 a), b). Note that our experiments use Adam on richer architectures while our theory analyses online SGD on a single-neuron denoiser. The qualitative agreement supports the robustness of the simplicity bias, but an exact quantitative match of timescales is not expected.

## 4.3. The importance of the spherical constraint for SGD

In order to consider a setting slightly closer to practice, we now remove the spherical constraint on SGD. The optimization algorithm then becomes simply:

$$\begin{cases} w_0 \sim \text{Unif}(\mathbb{S}^{d-1}) \\ w_{\tau+1} = w_\tau - \eta_d \nabla \mathscr{L}(w_\tau, x_\tau) \quad \tau > 1 \end{cases} \tag{18}$$

Surprisingly, this simple change can let performance of the denoiser greatly deteriorate, as we now discuss.

We can see this loss of performance most clearly in the single spike case with $\beta_u = 0$. Expanding the gradient of the population loss, we notice that the removal of the spherical constraint leads to appearance of an additional, radial term. Recalling the notation for the overlap with the hidden spiked direction $\alpha_\tau = v \cdot w_\tau$, we have:

$$-\nabla \mathcal{L}_t(w_\tau) = -\mathbb{E}\left[\nabla \mathscr{L}(w_\tau, x_\tau)\right] \tag{19}$$

$$= \underbrace{\left(\frac{c_{k^*}^L c_{k^*-1}^{\tilde{F}}}{(k^*-1)!}\alpha_\tau^{k^*-1} + O(\alpha_\tau^{k^*})\right) v}_{\text{signal term}} + \tag{20}$$

$$+ \underbrace{\left(c_1^{\tilde{F}} + c_0^{G_\sigma} + O(\alpha_\tau)\right) w_\tau}_{\text{additional radial term}}, \tag{21}$$

where $(c_i^{\tilde{F}})_{i\in\mathbb{N}}$ and $(c_j^G)_{j\in\mathbb{N}}$ are the Hermite coefficients of the functions:

$$\tilde{F}_\sigma(x_w, ||w||) = \sigma''(x_w)||w||^2 - \sigma'(x_w)\sigma(x_w)||w||^2 \\ - \sigma(x_w) - \sigma'(x_w)x_w,$$

$$G_\sigma(x_w) = 2\sigma'(x_w) - \sigma^2(x_w). \tag{22}$$

Due to the fact that we are not considering $w$ to have fixed norm, the coefficients $(c_i^{\tilde{F}})_{i\in\mathbb{N}}$ and $(c_j^G)_{j\in\mathbb{N}}$ depend on $||w||$ and are defined by the formula (see also section B.7 for more details on the expansion):

$$c_k^{\tilde{F}}(||w||) := \mathbb{E}_{x\sim\mathcal{N}(0,\mathbb{1})}\left[\partial^k \tilde{F}_\sigma(w \cdot x)\right] \tag{23}$$

$$c_k^G(||w||) = \mathbb{E}_{x\sim\mathcal{N}(0,\mathbb{1})}\left[\partial^k G_\sigma(w \cdot x)\right] \tag{24}$$

We can immediately see that the additional radial term strongly impacts the dynamics; in particular, in case $\sigma$ is an odd function (which implies $c_0^{\tilde{F}} = 0$) or $c_1^L = 0$, it is the leading contribution in eq. (19).

Zooming in on this case, many choices of $\sigma$ (especially if we pick $\sigma$ to be able to perfectly match the score, see fig. 2 and section 4.4) imply that $c_1^{\tilde{F}} + c_0^G < 0$, so it acts as a weight decay term that pushes $w_{\tau+1}$ towards zero. In case $c_2^L$ not large enough, the weight decay term overwhelms the signal term, leading to contraction dynamics that converge

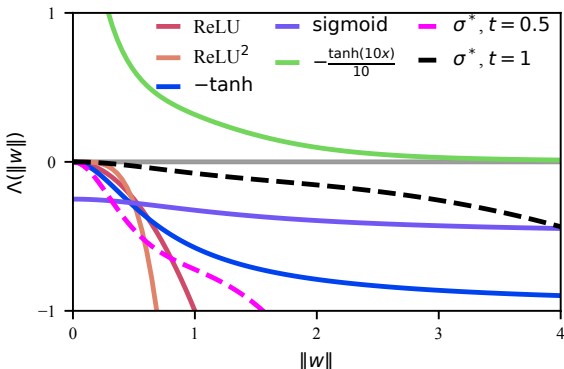

*Figure 2.* Examples of the contraction term $\Lambda$ for different choices of activation $\sigma$. $\sigma^*$ denotes the matched functional form of the score eq. (49) for different values of the diffusion time $t$.

to zero. In the next two paragraphs we will analyse the rigorous statement and interpret its implication.

**Proposition 4.7.** *In the setting described in section 4.3 with $c_1^L c_0^{\tilde{F}} = 0$ (for instance, this is verified when $\sigma$ is an odd function), with a population loss that can be expanded as eq. (19), with a $\mathcal{C}^\infty$ link function $\sigma$ such that $\Lambda(||w||) := (1 + c_2^L)c_1^{\tilde{F}}(||w||) + c_0^G$ and $\Lambda(||w||) \le -k_0 < 0$ for all $||w|| \le \Gamma$. Suppose we initialize such that $||w_0|| < \Gamma$ and $\alpha_0 \to 0$ as $d \to \infty$. We perform SGD as in eq. (18) with step size $\eta$, such that $\lim_{d\to\infty} \eta = 0$. Then there exists $\bar{d}$ such that for $d > \bar{d}$ the sequences $\alpha_\tau := w_\tau \cdot v$ and $||w_\tau||$ satisfy the following upper bound for all $\tau$ with probability that goes to 1 as $d \to \infty$:*

$$\alpha_\tau \le \bar{\gamma}^\tau \alpha_0 + r_d \tag{25}$$

$$||w_\tau|| \le \bar{\delta}^\tau ||w_0|| + s_d \tag{26}$$

*where $0 < \bar{\gamma}, \bar{\delta} < 1$ and $r_d, s_d \to 0$ as $d \to \infty$. In particular $\bar{d}$ is such that $\alpha_\tau \le \alpha_0$ and $||w_\tau|| \le ||w_0||$ for all $\tau \ge 0$.*

The proof is presented in section B.7 in the appendix.

Proposition 4.7 shows that for many choices of nonlinearity $\sigma$ SGD dynamics initialized isotropically with $w_0 \sim \text{Unif}(\mathbb{S}^{d-1})$ cannot escape an attracting minimum at $w = 0$: both the overlap $\alpha_\tau$ and the norm $||w_\tau||$ shrink to 0 without ever surpassing their value at initialization.

It may seem counterintuitive to see a phenomenology that is so different from the case of projected SGD discussed in propositions 4.3, 4.4 and 4.6. It turns out that this effect is due to precise properties of the data structure and the architecture that we are considering. First, we need to highlight the assumptions: $c_1^L c_0^{\tilde{F}} = 0$ means that data structures with *low information exponent*, $c_1^L \ne 0$ are less likely to show this behavior: in these cases the strong linear correlations in the data strongly push the gradient of the loss towards

non-trivial solutions, counterbalancing the attraction that the *additional radial term* in eq. (19) may present. In case of information exponent larger than one ($c_1^L = 0$), it is the assumption

$$\Lambda(||w||) = (1 + c_2^L)c_1^{\tilde{F}}(||w||) + c_0^G < 0 \qquad (27)$$

that regulates the attracting behavior of the fixed point $w = 0$. We can see in fig. 2 that it is possible to classify common activation functions based on this property (note that non-smooth functions such as ReLU or ReLU$^2$ violate the regularity assumptions of proposition 4.7, but it is still possible to work with smoothed versions of these nonlinearities). It is worth pointing out that in practice, it is possible to manually tweak $\sigma$ via small rescaling so that the sign of $\Lambda$ can change. An example is the case of $\sigma = -\tanh$: it is possible to consider a scaled version $\xi \to -\frac{\tanh(10\xi)}{10}$ (green line in fig. 2) for which $\Lambda$ is always positive. Note that the choice of the sign for $\sigma$ is driven by the necessity of requiring $C_3^{\tilde{F}_\sigma} C_4^L \geq 0$, otherwise the signal term in equation 19 would push in the wrong direction.

### 4.4. The case of the optimal denoiser

A particular case that is worth discussing in detail is the choice of $\sigma$ that could optimally represent the score of the data distribution (see the derivation of eq. (49) in the appendix). Recalling again the single spike setting with $\beta_u = 0$, for a fixed diffusion time $t$, we define the matched denoiser as

$$\sigma_t^*(\xi) = -\frac{e^{-t}}{\Delta_t}\left(e^{-t}\xi - \tanh\left(\frac{\xi e^{-t}}{\Delta_t}\right)\right). \qquad (28)$$

The denoiser $\sigma_t^*$ has the property that it is the function that can reach the smallest population loss value, attained for $w = v$, and its functional form can be derived by simply going through the steps in the derivation of eq. (5) backwards, see section B.3. Studying the optimization problem with $\sigma_t^*$ is the analog of studying denoising diffusion in the *teacher-student scenario*: a setting in which the architecture that needs to perform inference has the same structure of the data generating process, and only needs to infer the weights (see for instance Engel & Van den Broeck (2001) for a review).

In our setting, for all times $t \in (0, 1)$, $\sigma^*$ satisfies the assumptions of proposition 4.7 and the sub-optimal solution $w = 0$ is the effective attractor of the dynamics. This is the first example known to the authors of a model in which the teacher-student scenario is suboptimal with respect to the choice of the non-linearity. We can single out two factors that explain why the solution $w = 0$ is more stable than in other settings.

1. The most trivial contribution is given by the employment of MSE loss which provides the contraction term

$\sigma^2(x_w)$ inside $G_\sigma$. Even though this is an obvious contribution, it is worth pointing it out since many other theoretical works on single/multi index Gaussian models in supervised learning consider different loss functions, such as *correlation loss* studied in (Damian et al., 2023; Bardone & Goldt, 2024), that do not have this property.

2. The nature of the score denoising tasks. Indeed, part of the push towards 0 in the case of $\sigma^*$ is due to the term $2(\sigma^*)'(x_w)$ in $G_\sigma$. This term comes from integrating out the additional Gaussian noise present in equation 5 via Stein's lemma (lemma B.4), a specific contribution that is not present in usual supervised setting.

### 4.5. The role of overparametrization and depth

We investigated how more flexible architectures can learn higher-order statistics without task-specific tuning of the nonlinearity by training multi-layer autoencoders (matched to eq. (7)) and feed-forward residual networks on the MCM model (fig. 3, see section A.3 for experimental details). We measure learning by the maximal overlap $w^\top v/||w||$ between the first-layer weights and the spiked direction. We observe that over-parameterization can mitigate the curse predicted by proposition 4.7. In fig. 3, a wide two-layer autoencoder trained with Adam is able to learn the hidden direction $v$ with the activations ReLU and ReLU$^2$, even though $\Lambda < 0$ for both these nonlinearities, and we recall that $\Lambda < 0$ implies that training a single-neuron autoencoder with unprojected GD would get stuck near the trivial solution $w = 0$. Indeed, we show in fig. 7 that Adam applied to a narrow architecture remains stuck at zero. The over-parametrisation thus helps the network to escape the trivial solution. Over-parametrisation is well known to speed up learning in different settings, like learning two-layer ReLU networks (Safran et al., 2021) or phase retrieval (Sarao Mannelli et al., 2020). For the two-layer networks in fig. 7a), it could be that the repulsive interactions between neurons in wide networks (Mei et al., 2018) spread neurons apart and counteract the contraction toward $w = 0$ predicted by the single-neuron dynamics. Understanding this mechanism more precisely is an interesting direction for future work.

Width and depth, however, affect learnability in different ways. As depth increases, ReLU$^2$ networks lose the ability to recover the spike, likely due to exploding activations induced by repeated squaring. Strikingly, deeper networks with $\tanh$ or sigmoid activations do recover the spike. As discussed in section 4.3, simple reparametrizations can flip the sign of $\Lambda$, so that an activation violating the condition of proposition 4.7 can be transformed into an equivalent learnable representation. This suggests a possible role of depth: early layers can implement data-dependent transformations that effectively reparameterize the input seen by later layers,

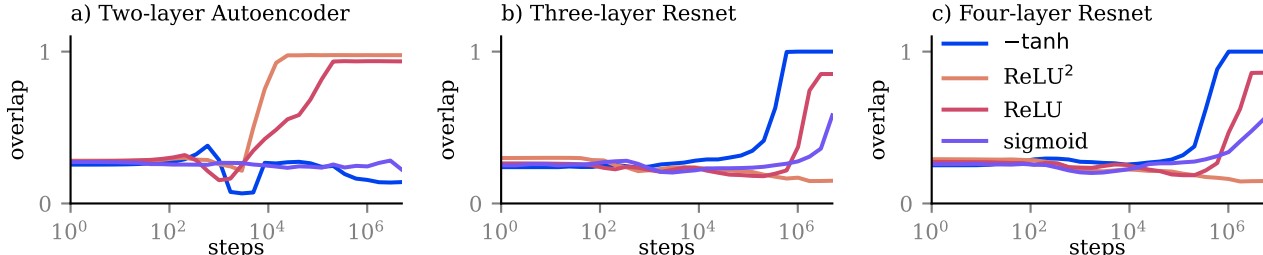

*Figure 3.* Normalized overlap of first-layer weights of neural networks of varying depth trained with Adam on inputs drawn from the mixed cumulant model, eq. (6), at $d = 100$. All curves are averages over 5 random initializations of the neural networks.

improving the conditioning of the nonlinearity and thereby restoring learnability without explicit tuning of $\sigma$.

## 5. Concluding perspectives

We demonstrated experimentally that diffusion models learn statistics of increasing complexity over training, where complexity is defined via the order of the cumulants exploited by the model. In the mixed cumulant model, we rigorously analyze this phenomenon and show that learning is governed by the diffusion information exponent $k^*$, which determines the sample complexity required to recover a given statistic and depends on both the architecture and the data distribution; correlated latent variables facilitate learning. While our theoretical analysis focuses on a simple denoiser with a single hidden unit and a simplified data distribution with one or two non-Gaussian directions, the diffusion information exponent can in principle be computed for richer architectures and more structured data models.

We further compare two optimization protocols and show that SGD projected on the sphere outperforms its unconstrained counterpart, which can be trapped by a suboptimal attracting solution at $\|w\| = 0$ for several common nonlinearities. Numerical experiments indicate that increasing network width and depth restores the ability to learn higher-order statistics, suggesting that both overparametrization and depth can mitigate unfavorable nonlinearities without explicit fine-tuning. Overall, our results show that diffusion models exhibit a distributional simplicity bias beyond supervised learning: when data or model expressivity is limited, they reliably learn lower-order statistics before higher-order ones.

## Code availability

The code to reproduce our experiments is available at https://github.com/ClaudiaMer/DiffusionEasy2Hard.

## Acknowledgements

SG and CM gratefully acknowledge funding from the European Research Council (ERC) for the project "beyond2", ID 101166056. SG also acknowledges funding from the European Union–NextGenerationEU, in the framework of the PRIN Project SELF-MADE (code 2022E3WYTY – CUP G53D23000780001), and from Next Generation EU, in the context of the National Recovery and Resilience Plan, Investment PE1 – Project FAIR "Future Artificial Intelligence Research" (CUP G53C22000440006).

## Impact statement

This paper presents work whose goal is to advance the field of Machine Learning. There are many potential societal consequences of our work, none which we feel must be specifically highlighted here.

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

a) mean b) mean + cov. c) test

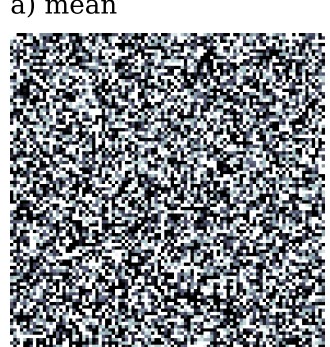 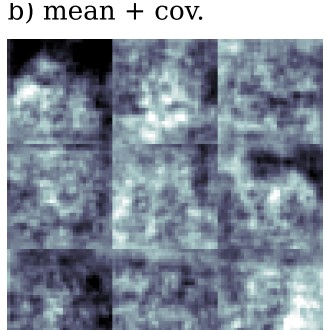 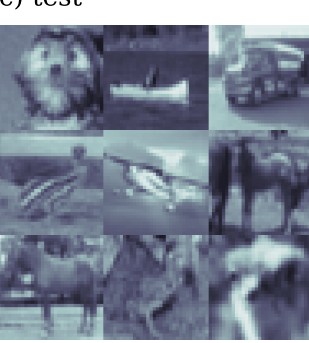

*Figure 4.* Samples from the different "clones" as well as the test data set. a) shows images drawn from the mean clone which follows a Gaussian distribution with matching mean to the CIFAR-10 dataset and identity covariance. In b), we additionally match the covariance matrix of the Gaussian distribution to the CIFAR-10 dataset. c) shows 9 images from the CIFAR-10 dataset.

a) mean b) mean + cov. c) test

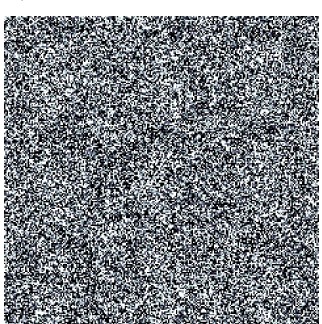 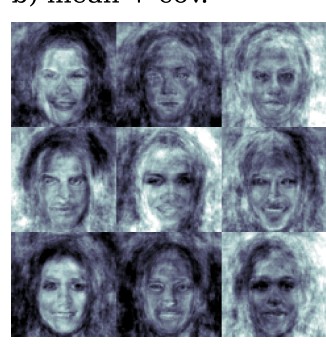 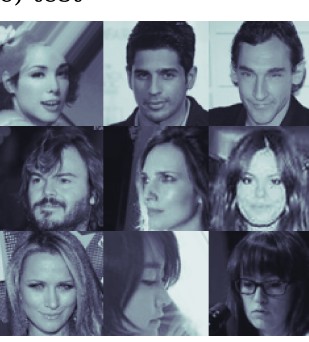

*Figure 5.* Samples from the different "clones" as well as the test data set. a) shows images drawn from the mean clone which follows a Gaussian distribution with matching mean to the CelebA dataset and identity covariance. In b), we additionally match the covariance matrix of the Gaussian distribution to the CelebA dataset. c) shows 9 images from the CelebA dataset.

## A. Details on experiments

### A.1. The clones

To generate the clone datasets, we first determine the mean $\mu$ and the covariance $\Sigma$ of the test datasets for both CelebA and CIFAR-10. We then sample the "mean" clone from a Gaussian distribution with mean $\mu$ and identity covariance. We then sample the "mean + cov." clone from a Gaussian distribution with matching mean and covariance. We show examples of these datasets in fig. 4 and fig. 5.

### A.2. Sequential learning in CelebA data

In fig. 6 we report the outcome of the sequential learning experiment on $10^5$ CelebA data (Liu et al., 2015), which we downscale to $80 \times 80$ greyscale pixels. We observe the same sequential learning behavior as for the CIFAR-10 data. We observe that the loss curves are more noisy than the ones for CIFAR-10, we expect these effects to be due to the finite learning rate used during training.

#### A.2.1. TRAINING HYPERPARAMETERS

We use diffusion models with a Unet architecture (Ronneberger et al., 2015), $T = 10^3$ levels of noise and sinusoidal embedding for $t$. For the CIFAR-10 data reported in fig. 1, we use the Adam optimizer with learning rate $10^-3$. For the CelebA data reported in fig. 6, we reduce the learning rate to $10^{-4}$. For both data sets we use a batchsize of $10^2$ samples.

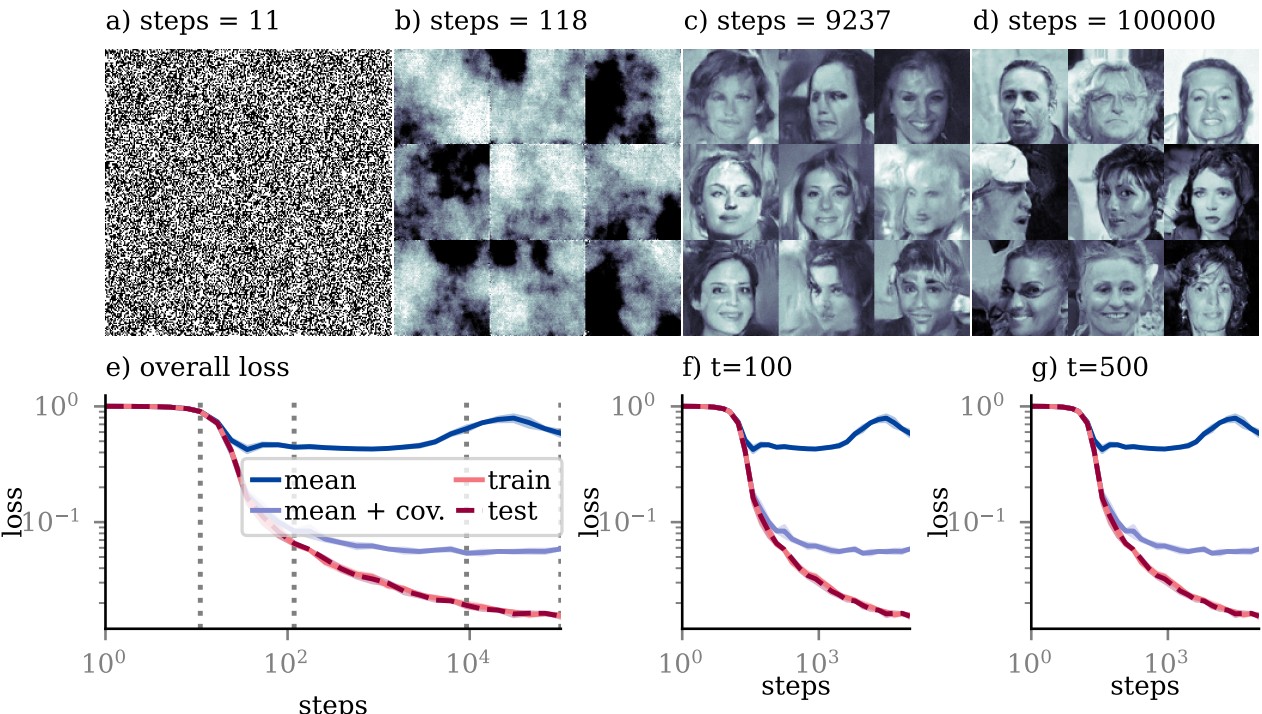

*Figure 6.* Training Unets on CelebA data. a)-d) samples generated from the model at various training stages. e) Test loss of the model on clones of the dataset during training. Vertical dotted lines mark training stages of images generated from the model shown in panels a)-d). f)-g) same as e), but for fixed level of noise. All curves are averaged over 2 initializations of the network models and $5 \cdot 10^3$ test data. Shaded areas report standard deviation over random initialization.

### A.3. Learning the MCM model

In this section, we describe the different architectures trained on the MCM model. We sort these descriptions by the figures they correspond to.

**fig. 1c): $n$-layer feedforward network with linear skip connection.** Let $x \in \mathbb{R}^d$ be the input drawn according to eq. (6) with $\beta_u = 10^2, \beta_v = 1$. The network defines a sequence of hidden representations $\{h^{(k)}\}_{k=1}^{d-1} \subset \mathbb{R}^m$ as follows:

$$h^{(1)} = g(W_1 x), \tag{29}$$

$$h^{(k)} = g\left(W_k h^{(k-1)}\right), \qquad k = 2, \ldots, n-1, \tag{30}$$

where $W_1 \in \mathbb{R}^{m \times d}, W_k \in \mathbb{R}^{m \times m}$ for $k \geq 2$ are learnable weight matrices, $g(\cdot)$ is an element-wise nonlinearity, and $\lambda \in \mathbb{R}$ is a fixed residual scaling parameter, which we set to one. The output of the network is given by

$$f(x) = \alpha x + W_{\text{out}} h^{(n-1)} + Ax + b, \tag{31}$$

where $W_{\text{out}} \in \mathbb{R}^{d \times m}$ is a learnable output weight matrix and $\alpha \in \mathbb{R}$ is a fixed skip-connection coefficient. We have also added a general linear mapping $Ax + b$ to the output to allow the network to learn the Gaussian part of the score explicitly; $A \in \mathbb{R}^{d \times d}, b \in \mathbb{R}^d$ is a learnable matrix and bias vector. This architecture was used to produce panel c) fig. 1, choosing $\sigma =$ReLu and $m = 10, n = 3$. To train it, we used objective eq. (5), the Adam optimizer with learning rate $\eta = 10^{-3}$ and a batch-size of $10^2$ data samples per step.

**fig. 3 a): Twolayer Autoencoder.** We consider a two-layer neural network with tied weights. Given an input $x \in \mathbb{R}^d$ drawn according to eq. (6) with $\beta_u = 0, \beta_v = 1$, the hidden representation is computed as

$$h = \sigma(Wx)$$

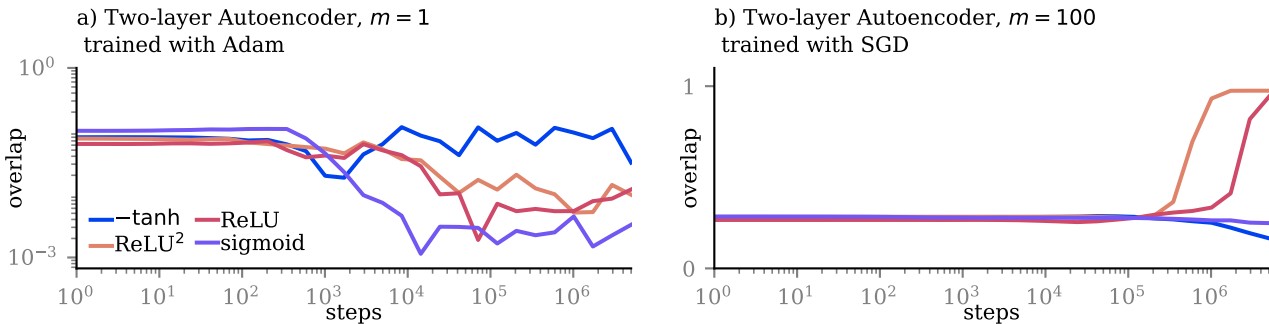

*Figure 7.* Normalized overlap of first-layer weights of neural networks of varying $m$ trained with Adam or SGD the MCM model at $d = 100$. All curves are averages over 5 random initializations of the neural networks.

where $W \in \mathbb{R}^{m \times d}$ and $\sigma(x)$ is an element-wise nonlinearity. We also add a trainable skip connection $\alpha$, so the output is given by

$$S(x) = -\alpha x - W^{\mathrm{T}} h + b \tag{32}$$

where the second layer reuses the transpose of the first-layer weights and $b \in \mathbb{R}^d$ is a learnable bias. This architecture was used to produce panel a) fig. 3, choosing different activation functions for $\sigma$. To train it, we used objective eq. (5), the Adam optimizer with learning rate $\eta = 10^{-4}$ and a batch-size of $10^2$ data samples per step.

**fig. 3 b-c):** $n$-**layer Resnet**. Let $x \in \mathbb{R}^d$ be the input drawn according to eq. (6) with $\beta_u = 0, \beta_v = 1$. The network defines a sequence of hidden representations $\{h^{(k)}\}_{k=1}^{d-1} \subset \mathbb{R}^m$ as follows:

$$h^{(1)} = g(W_1 x), \tag{33}$$

$$h^{(k)} = \lambda\, h^{(k-1)} + g\left(W_k h^{(k-1)}\right), \qquad k = 2, \ldots, n-1, \tag{34}$$

where $W_1 \in \mathbb{R}^{m \times d}$, $W_k \in \mathbb{R}^{m \times m}$ for $k \geq 2$ are learnable weight matrices, $g(\cdot)$ is an element-wise nonlinearity, and $\lambda \in \mathbb{R}$ is a fixed residual scaling parameter, which we set to one. The output of the network is given by

$$f(x) = \alpha x + W_{\mathrm{out}}\, h^{(n-1)}, \tag{35}$$

where $W_{\mathrm{out}} \in \mathbb{R}^{d \times m}$ is a learnable output weight matrix and $\alpha \in \mathbb{R}$ is a fixed skip-connection coefficient. All linear layers are bias-free. This architecture was used to produce panel b)-c) fig. 3, choosing different activation functions for $\sigma$. To train it, we used objective eq. (5), the Adam optimizer with learning rate $\eta = 10^{-4}$ and a batch-size of $10^2$ data samples per step.

### A.4. Further experiments on learning the MCM model

Here we present two further experiments on the MCM model using the same architecture as in fig. 3a), see section A.3 for details. We modify the setting used to obtain fig. 3a) in two ways: first, we remove the overparametrization, i.e., we set the width of the autoencoder to $m = 1$. We find that the model can no longer recover the spike. In the second experiment, we keep the width of the autoencoder at $m = 100$, meaning that the network is overparametrized, but use SGD instead of Adam as an optimization regime. Further, we increase the learning rate to $\eta = 10^{-2}$, as SGD can have slower convergence than Adam. We find that using SGD, the same architectures are able to recover the spike as in the case where we trained with Adam. We hence conclude that overparametrization is the key for recovery of the spike.

## B. Details on the analysis and proofs

### B.1. Notation

We made use of asymptotic notation as follows. For two positive functions $f$ and $g$ from $\mathbb{N} \to \mathbb{R}$ (we recall the definitions in the discrete setting, but they trivially extend also for functions defined on $\mathbb{R}$), we write $f(k) = O(g(k))$ if there exist constants $C > 0$ and $k_0$ such that $f(k) \leq Cg(k)$ for all $k \geq k_0$. We write $f(k) = o(g(k))$ if $f(k)/g(k) \to 0$ as $k \to \infty$. Analogously, $f(k) = \Omega(g(k))$ if there exists constants $c > 0$ and $k_0$ such that $f(k) \geq cg(k)$ for all $k \geq k_0$,

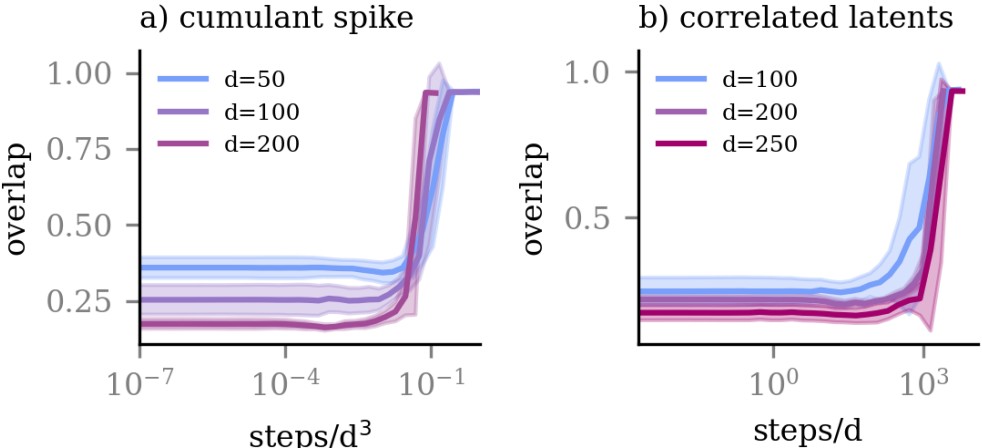

*Figure 8.* Normalized overlap of first-layer weights with cumulant spike of neural networks trained with SGD on inputs drawn from the mixed cumulant model over training steps, divided by the (cubed) dimension $d$. a) shows the case of a single cumulant spike, with number of steps scaled by $d^{-3}$. Panel b) shows the case of an MCM model with a covariance and a cumulant spike, which have correlated latents, with training steps scaled by $d^{-1}$. All curves are averages over 5 random initializations of the neural networks, with two-layer autoencoder architecture and ReLu nonlinearity in the overparametrized regime.

while $f(k) = \omega(g(k))$ if $f(k)/g(k) \to \infty$ as $k \to \infty$. Finally, $f(k) = \Theta(g(k))$ means that both $f(k) = O(g(k))$ and $f(k) = \Omega(g(k))$ hold simultaneosly.

## B.2. Hermite polynomials

We recall the definition and a few properties of the Hermite polynomials.

**Definition B.1.** The Hermite polynomial of degree $m$ is

$$h_m(x) := (-1)^m e^{\frac{x^2}{2}} \frac{\mathrm{d}^m}{\mathrm{d}x^m} \left( e^{-\frac{x^2}{2}} \right) \tag{36}$$

There is also a general formula:

$$h_m(x) = m! \sum_{j=0}^{\lfloor m/2 \rfloor} \frac{(-1)^j}{2^j j! (m-2j)!} x^{m-2j} \tag{37}$$

The Hermite polynomials enjoy the following properties (for details see McCullagh (2018), Szegő (1939) and Abramowitz & Stegun (1964)):

- they are an orthogonal system with respect to the $L^2$ product weighted with the density of the Normal distribution:

$$\frac{1}{\sqrt{2\pi}} \int_{-\infty}^{\infty} h_n(x) h_m(x) e^{-\frac{x^2}{2}} \mathrm{d}x = n! \delta_{m,n}; \tag{38}$$

- $h_m$ is a monic polynomial of degree $m$, hence $(h_m)_{m \in \{1,...,N\}}$ generates the space of polynomials of degree $\leq N$;

- the previous two properties imply that the family of Hermite polynomials is an orthogonal basis for the Hilbert space $L^2(\mathbb{R}, \mathbb{Q})$ where $\mathbb{Q}$ is the normal distribution;

- they enjoy the following recurring relationship

$$h_{m+1}(x) = x h_m(x) - h'_m(x) = x h_m(x) - m h_{m-1}(x), \tag{39}$$

**Multivariate case**  In the multivariate $m$-dimensional case we can generalize to Hermite tensors $(H_{\boldsymbol{\alpha}})_{\boldsymbol{\alpha} \in \mathbb{N}^m}$ defined as:

$$H_\alpha(x_1, \ldots, x_m) = \prod_{i=1}^{m} h_{\alpha_i}(x_i) \tag{40}$$

most of the properties of the one-dimensional Hermite polynomials extend to this case: they form an orthogonal basis of $L^2(\mathbb{R}^m, \mathcal{N}(0, \mathbb{1}_m))$. We have that:

$$\mathbb{E}_{x \sim \mathcal{N}(0,\mathbb{1})}[H_{\boldsymbol{\alpha}}(x) H_{\boldsymbol{\beta}}(x)] = \boldsymbol{\alpha}! \delta_{\boldsymbol{\alpha},\boldsymbol{\beta}} \tag{41}$$

**Definition B.2** (Hermite expansion).  Consider a function $f : \mathbb{R} \to \mathbb{R}$ that is square integrable with weight the standard normal distribution $p(x) = (1/\sqrt{2\pi}) \, e^{-x^2/2}$. Then, there exists a unique sequence of real numbers $\{c_k\}_{k \in \mathbb{N}}$ called Hermite coefficients, such that:

$$f(x) = \sum_{k=0}^{\infty} \frac{c_k}{k!} h_k(x) \quad \text{and} \quad c_k := \mathbb{E}_{x \sim \mathcal{N}(0,1)}[f(x) h_k(x)],$$

where $h_i$ is the $i$-th probabilist's Hermite polynomial.

**Definition B.3** (Information exponent).  Consider a function $f$ that can be expanded with Hermite polynomials with coefficients $(c_i)_i \in \mathbb{N}$. Its information exponent $k^* = k^*(f)$ is the smallest index $k \geq 1$ such that $c_k \neq 0$.

The following lemma from (Bandeira et al., 2020) provides a version of the integration by parts technique that is tailored for Hermite polynomials.

**Lemma B.4** (Stein lemma).  *Let $f : \mathbb{R}^d \to \mathbb{R}^d$ be a continuously differentiable $k$ times function. Suppose that $f$ and all of its partial derivatives up to the $k$-th are bounded by $O(\exp(|y|^\lambda))$ for a $\lambda \in (0, 2)$, then for any $\boldsymbol{\alpha} \in \mathbb{N}^d$ such that $|\boldsymbol{\alpha}| \leq k$*

$$\mathbb{E}_{y \sim \mathcal{N}(0,\mathbb{1}_d)}[H_{\boldsymbol{\alpha}}(y) f(y)] = \mathbb{E}_{y \sim \mathcal{N}(0,\mathbb{1}_d)}[\partial_{\boldsymbol{\alpha}} f(y)] \tag{42}$$

*Proof.*  42 can be proved by doing induction on $k$ using 39, see (Bandeira et al., 2020) for details. □

**Corollary B.5.**  *Let $u_1, u_2 \in S^{d-1}$, then the following formula holds:*

$$\mathbb{E}_{x \sim \mathcal{N}(0,\mathbb{1}_d)}[h_i(u_1 \cdot x) h_j(u_2 \cdot x)] = (u_1 \cdot u_2)^i i! \delta_{i,j} \tag{43}$$

*Proof.*  It follows from the application of lemma B.4 □

### B.3. Derivation of the loss formula

We consider the setting of (Biroli & Mézard, 2023), modeling the diffusion process for $t \to x(t) \in \mathbb{R}^d$, with $x(0) = a \sim P_0$, the target distribution, that we want to learn how to sample from. The diffusion process has the following form:

$$\mathrm{d}x(t) = -x\mathrm{d}t + \mathrm{d}\mathcal{W}_t, \tag{44}$$

where $\mathcal{W}_t \in \mathbb{R}^d$ is a $d$ dimensional Wiener process. The solution at time $t$ can be written in distribution as:

$$x(t) = x(0)e^{-t} + \sqrt{1 - e^{-2t}} z \tag{45}$$

where $z \sim \mathcal{N}(0, \mathbb{1}_d)$. So, defining $\Delta_t = 1 - e^{-2t}$ The density at time $t$ is given by:

$$P_t(x) = \int \mathrm{d}a \, P_0(a) \frac{1}{(2\pi\Delta_t)^{d/2}} \exp\left(-\frac{1}{2} \frac{(x - ae^{-t})^2}{\Delta_t}\right). \tag{46}$$

A key quantity that needs to be introduced is the *score* of $P_t$, that appears in the equation for the backward process. Knowing the score, or being able to approximate it, allows to revert the process and, starting from samples of a standard Gaussian it gives the recipe on how to transform them into samples of $P_0$. The score is defined as:

$$\mathcal{F}_i(x, t) = \frac{\partial \log P_t(x)}{\partial x_i} = -\frac{x_i - \mathbb{E}[a_i | x(t) = x] e^{-t}}{\Delta_t}, \tag{47}$$

where the last equality, called *Tweedie formula*, is at the core of the feasibility of diffusion models. It gives a recipe on how to approximate the score via empirical averages of the noised process. The objective then becomes learning $\mathcal{F}$. To do this, one can build MSE for a collection of fixed time intervals, let us denote $\mathcal{S}_t^w(x)$ the approximated score that depends on weight $w$. The loss function at time $t$ will be

$$
\mathcal{L}_t(w) = \frac{1}{2}\mathbb{E}_x\left[\left\|\mathcal{S}_t^w(x) + \frac{x - \mathbb{E}[a|x(t) = x]e^{-t}}{\Delta_t}\right\|^2\right]
$$

$$
\overset{eq.\,(45)}{=} \frac{1}{2}\mathbb{E}_{a,z}\left[\left\|\mathcal{S}_t^w(x) + \frac{ae^{-t} + \sqrt{\Delta}z - \mathbb{E}[a|x(t) = x]e^{-t}}{\Delta_t}\right\|^2\right]
$$

$$
= \frac{1}{2}\mathbb{E}_{a,z}\left[\left\|\mathcal{S}_t^w(x) + \frac{z}{\sqrt{\Delta_t}}\right\|^2 + \left\|\frac{ae^{-t} - \mathbb{E}[a|x(t) = x]e^{-t}}{\Delta_t}\right\|^2 + 2\frac{z}{\sqrt{\Delta_t}} \cdot \frac{ae^{-t} - \mathbb{E}[a|x(t) = x]e^{-t}}{\Delta_t}\right],
$$

where the double product with $\mathcal{S}_t^w$ does not appear because $\mathbb{E}_x[\mathcal{S}_t^w(x)\mathbb{E}_{a|x}[(a - \mathbb{E}[a|x(t) = x])]] = 0$. Note that the second and third addend are independent of $w$, hence the troublesome term $\mathbb{E}[a|x(t) = x]$ does not appear in the gradient and the effective loss function is:

$$
L_t(w) = \frac{1}{2}\mathbb{E}_{a,z}\left[\left\|\mathcal{S}_t^w(ae^{-t} + \sqrt{\Delta_t}z) + \frac{z}{\sqrt{\Delta_t}}\right\|^2\right] + C. \tag{48}
$$

This quantity can be well approximated having just samples from $P_0$, which allow to estimate the integral over $a$.

### B.4. The example of the spiked cumulant model

To provide intuition on the forward and backward processes described in section 4, we provide here the explicit formulas in the case in which $P_0$ is the spiked cumulant model from (Székely et al., 2024). We can choose the case $x(0) = \nu v + z$ with $\nu \sim \text{Rademacher}(1/2)$, $v$ is the norm 1 spike and $z$ is a $d-1$-standard Gaussian, in the space orthogonal to $v$, i.e. $z \sim \mathcal{N}(0, \mathbb{1}_d - vv^\top)$. Then from eq. (46), $P_t$ has the following expression:

$$
P_t(x) = \frac{1}{(2\pi)^{(d-1)/2}}\exp\left(-\frac{1}{2}x_{\perp v}^\top x_{\perp v}\right)\frac{1}{2(2\pi\Delta_t)^{1/2}}\left[\exp\left(-\frac{1}{2}\frac{(x_v - e^{-t})^2}{\Delta_t}\right) + \exp\left(-\frac{1}{2}\frac{(x_v + e^{-t})^2}{\Delta_t}\right)\right]
$$

$$
= \frac{1}{(2\pi)^{(d-1)/2}}\exp\left(-\frac{1}{2}x_{\perp v}^\top x_{\perp v}\right)\frac{\exp\left(-\frac{1}{2}\frac{x_v^2 + e^{-2t}}{\Delta_t}\right)}{(2\pi\Delta_t)^{1/2}}\cosh\left(\frac{x_v e^{-t}}{\Delta_t}\right),
$$

where $x = x \cdot vv + (x - x \cdot vv) = x_v v + x_{\perp v}$.

Hence the score is:

$$
\mathcal{F}(x,t) = -x_{\perp v} - \frac{x_v}{\Delta_t}v + \frac{e^{-t}}{\Delta_t}v\tanh\left(\frac{x_v e^{-t}}{\Delta_t}\right)
$$

$$
= -x - \frac{e^{-t}}{\Delta_t}v\left(e^{-t}x_v - \tanh\left(\frac{x_v e^{-t}}{\Delta_t}\right)\right). \tag{49}
$$

### B.5. Projected SGD dynamics

We will now focus on the dynamics of projected SGD, so we can assume $||w|| = 1$. Denoting for brevity by $x_w := x \cdot w$:

$$
-\nabla_{sph}\mathcal{L}_t = -(\mathbb{1} - ww^\top)\mathbb{E}_{x,z}\Bigg[\sigma'(x_w)\sigma(x_w)x + \sigma^2(x_w)w + \sigma(x_w)x +
$$

$$
+ \sigma'(x_w)x_w x - \frac{1}{\sqrt{\Delta_t}}\left(\sigma(x_w)z + \sigma'(x_w)z_w x\right)\Bigg]
$$

$$
= (\mathbb{1} - ww^\top)\mathbb{E}_x\left[x\left(\sigma''(x_w) - \sigma'(x_w)\sigma(x_w) - \sigma(x_w) - \sigma'(x_w)x_w\right)\right]
$$

where the second equality all the terms proportional to $w$ have been canceled by the factor $\mathbb{1} - ww^\top$ and the terms depending on $z$ can be reduced to terms involving just $x$ through Stein lemma (as detailed in lemma F.1 in (Shah et al., 2023)):

$$\mathbb{E}_{z \sim \mathcal{N}(0, \mathbb{1}), x} \left[ \frac{1}{\sqrt{\Delta_t}} \sigma(x_w) z \right] = \mathbb{E}_x \left[ \sigma'(x_w) w \right]$$

$$\mathbb{E}_{z \sim \mathcal{N}(0, \mathbb{1}), x} \left[ \frac{1}{\sqrt{\Delta_t}} \sigma'(x_w) z_w x \right] = \mathbb{E}_x \left[ \sigma''(x_w) ||w||^2 x + \sigma'(x_w) w \right]$$

Then introducing $L(v \cdot x)$ and defining

$$F_\sigma(x_w) := \sigma''(x_w) - \sigma'(x_w)\sigma(x_w) - \sigma(x_w) - \sigma'(x_w)x_w$$

we can expand in Hermite orthonormal basis:

$$L(v \cdot x) = \sum_{i=0}^{\infty} c_i^L h_i(x_v)$$

$$F_\sigma(x_w) = \sum_{j=0}^{\infty} c_j^F h_j(x_w)$$

and get:

$$-\nabla_{sph}\mathcal{L}_t = (\mathbb{1} - ww^\top) \underset{x \sim \mathcal{N}(0, \mathbb{1})}{\mathbb{E}} \left[ x \left( \sum_{i=0}^{\infty} c_i^L h_i(x_v) \right) \left( \sum_{j=0}^{\infty} c_j^F h_j(x_w) \right) \right]$$

$$= (\mathbb{1} - ww^\top) \left[ \left( \sum_{i=1}^{\infty} c_i^L c_{i-1}^F (v \cdot w)^{i-1} \right) v + \left( \sum_{j=1}^{\infty} c_j^F c_{j-1}^L (v \cdot w)^{j-1} \right) w \right]$$

$$= \left( \sum_{i=1}^{\infty} c_i^L c_{i-1}^F (v \cdot w)^{i-1} \right) (1 - v \cdot w) v$$

So, let $\alpha := v \cdot w$ in the early stages of learning the dynamics to reach weak recovery are described by:

$$-\nabla_{sph}\mathcal{L}_t = c_{k^*}^L c_{k^*-1}^F \alpha^{k^*-1} v + O(\alpha^{k^*}) \tag{50}$$

where $k^*$ is the first non zero term of the series. In the following we consider some examples of applications of proposition 4.3.

*Proof of proposition 4.3 and proposition 4.4.* These propositions can be seen as corollaries of theorems 1.3 and 1.4 from (Ben Arous et al., 2021). Assumptions 4.1 and 4.2 verify the core requirements. The only catch is that all the assumptions are not verified on the loss function, but directly on the spherical gradient. However the whole reasoning detailed in (Ben Arous et al., 2021) never relies on computations of the loss function, but only of its gradient, hence we can apply the proof to our setting. □

### B.5.1. SPIKED WISHART

In the case of spiked Wishart model, $k^* = 2$, so we can take $\sigma = -id$:

$$F(x_w) = x_w$$

and we get that the projected SGD reaches weak recovery in $d \log d$ sample complexity

### B.5.2. SPIKED CUMULANT

We choose $\sigma$ to match equation 49:

$$\sigma(x_v) = \frac{e^{-t}}{\Delta_t} \left( e^{-t} x_v - \tanh\left( \frac{e^{-t}}{\Delta_t} x_v \right) \right)$$

and simulations confirm that projected SGD works in this regime reaching weak recovery in $d^3$ samples. Note that the coefficients depend exponentially on diffusion time $c_4^L = e^{-4t} - 3e^{-2t}$

### B.6. Mixed cumulant model

*Proof of proposition 4.6.* The proof relies on verifying the precise hypothesis of propositions 3 and 4 in (Bardone & Goldt, 2024), so that all the argument can be replicated in the exact same way. The starting point is that the population loss expansion

$$-\nabla_{\text{sph}}\mathcal{L}(\alpha_u, \alpha_v) = \sum_{k=1}^{\infty}\sum_{i=0}^{k} c_{k-1}^F c_{i,k-i}^L \left(\alpha_u^{i-1}\alpha_v^{k-i}u + \alpha_u^i \alpha_v^{k-i-1}v\right) \tag{51}$$

coincides with the population loss from (Bardone & Goldt, 2024), eq. (25), with a different naming of the coefficients: in their notation $k c_k^\sigma$ corresponds to $c_{k-1}^F$ in our notation. Hence assumption 4.5 verifies the requirements of Assumption 1 in (Bardone & Goldt, 2024). The only term that in principle could behave differently is the directional noise martingale $H_d(x, w) := \mathscr{L} - \mathcal{L}$. However $L_t$ is the likelihood ratio of a sub-Gaussian random variable, and $H$ is a Lipschitz transformation, so $H(x, w)$, with $||w|| = 1$ and $x \sim \mathbb{P}_t$ is sub-Gaussian. Hence requirements in assumption 4.2, which were the same as the ones needed in (Bardone & Goldt, 2024), are satisfied. Hence, we can apply propositions 3-4 from (Bardone & Goldt, 2024) and conclude the proof. $\square$

### B.7. Additional details for SGD without projection

We now do not consider the restriction to $||w|| = 1$ and the spherical gradient, SGD gradient has the form

$$-\nabla \mathcal{L}_t = -\mathbb{E}_{x,z}\left[\sigma'(x_w)\sigma(x_w)||w||^2 x + \sigma^2(x_w)w + \sigma(x_w)x + \sigma'(x_w)x_w x - \frac{1}{\sqrt{\Delta_t}}\left(\sigma(x_w)z + \sigma'(x_w)z_w x\right)\right]$$

$$= \mathbb{E}_x\left[x\left(\underbrace{\sigma''(x_w)||w||^2 - \sigma'(x_w)\sigma(x_w)||w||^2 - \sigma(x_w) - \sigma'(x_w)x_w}_{\tilde{F}_\sigma}\right) + w\left(\underbrace{2\sigma'(x_w) - \sigma^2(x_w)}_{G_\sigma}\right)\right]$$

More in general we can now expand the likelihood ratio $L, \tilde{F}_\sigma$ and $G_\sigma$ in Hermite basis. Note that now $||w||$ is not anymore constant equal to 1, as in the projected SGD case, so we will expand with respect to $\hat{w} = w/||w||$ and $v$ to get:

$$-\nabla \mathcal{L}_t = \mathbb{E}_x\left[x\tilde{F}_\sigma(x_w, ||w||) + wG_\sigma(x_w)\right]$$

$$= \mathop{\mathbb{E}}_{x\sim\mathcal{N}(0,\mathbb{1})}\left[x\left(\sum_{i=0}^{\infty}\frac{c_i^L}{i!}h_i(x_v)\right)\left(\sum_{j=0}^{\infty}\frac{c_j^{\tilde{F}}(||w||)}{j!}||w||^j h_j(x_{\hat{w}})\right) + \right.$$

$$\left. + w\left(\sum_{i=0}^{\infty}\frac{c_i^L}{i!}h_i(x_v)\right)\left(\sum_{k=0}^{\infty}\frac{c_k^G(||w||)}{k!}||w||^k h_k(x_{\hat{w}})\right)\right]$$

$$= \left[\left(\sum_{i=1}^{\infty}\frac{c_i^L c_{i-1}^{\tilde{F}}}{(i-1)!}(v\cdot w)^{i-1}\right)v + \left(\sum_{j=1}^{\infty}\frac{c_j^{\tilde{F}} c_{j-1}^L}{(j-1)!}(v\cdot w)^{j-1} + \sum_{k=0}^{\infty}\frac{c_k^G c_k^L}{k!}(v\cdot w)^k\right)w\right]$$

Where we applied Stein's lemma multiple times and the Hermite decomposition. The coefficients $(c_i^{\tilde{F}})_{i\in\mathbb{N}}$ and $(c_j^G)_{j\in\mathbb{N}}$ depend on $||w||$ and are defined as:

$$||w||^k c_k^{\tilde{F}}(||w||) = \mathbb{E}[\tilde{F}_\sigma(w\cdot x)h_k(\hat{w}\cdot x)] \overset{Stein}{=} ||w||^k \mathbb{E}\left[\partial^k \tilde{F}_\sigma(w\cdot x)\right] \tag{52}$$

$$||w||^k c_k^G(||w||) = \mathbb{E}[G_\sigma(w\cdot x)h_k(\hat{w}\cdot x)] \overset{Stein}{=} ||w||^k \mathbb{E}\left[\partial^k G_\sigma(w\cdot x)\right] \tag{53}$$

We now approximate at leading order in $\alpha := w\cdot v$, which at the beginning of learning is small. Recalling that $k^*$ denotes the diffusion information exponent, and $c_0^L = 1$, we get

$$-\nabla \mathcal{L}_t(w) = \left(c_1^{\tilde{F}} + c_0^G\right)w + O(\alpha)w + \frac{c_{k^*}^L c_{k^*-1}^{\tilde{F}}\alpha^{k^*-1}}{(k-1)!}v + O(\alpha^{k^*}) \tag{54}$$

So the dynamics at leading order are governed by $\Lambda := c_1^{\tilde{F}}(||w||)(1 + c_2^L) + c_0^G$ (note that the coefficient of $c_1^{\tilde{F}}$ comes from both terms of eq. (54) ). If $\Lambda < 0$ in a neighborhood of 0, then $w = 0$ is an attracting fixed point meaning that this analysis

is already able to characterize GD dynamics starting from random initializations (provided the basin of attraction includes the initialization), otherwise if $\Lambda > 0$ it is a repulsive fixed point, and the dynamics push away from the region in which it is possible to expand the loss, hence analysis of this case cannot be done through this method.

*Proof of proposition 4.7.* An iteration of the SGD can be written as:

$$w_{\tau+1} = w_\tau - \eta \left(\nabla \mathcal{L}(w_\tau) + \nabla H(w_\tau, x_\tau)\right)$$

using the expansion eq. (19) we get

$$\begin{cases} \alpha_{\tau+1} = \alpha_\tau + \eta \left(\alpha_\tau \Lambda(||w_\tau||) + E_\tau \alpha_\tau + v \cdot \nabla H(w_\tau, x_\tau)\right) \\ w_{\tau+1} = (1 + \eta \Lambda(||w_\tau||))w_\tau + \eta \vec{R}_\tau + \eta \nabla H(w_\tau, x_\tau) \end{cases} \tag{55}$$

where $|E_\tau| \leq L\alpha_\tau$ for some constant $L$ and $\vec{R}_\tau$ is a vectorized analogous from which we do not factor $\alpha_\tau$: $\left\|\vec{R}_\tau\right\| \leq L\alpha$. Now call $\gamma_\tau := 1 + \eta \Lambda(||w_\tau||) + \eta E_\tau$ and $\delta_\tau = 1 + \eta \Lambda(||w_\tau||)$, then it can be easily verified by induction that the recursive relations equation 55 implies the following explicit formulas:

$$\begin{cases} \alpha_\tau = \alpha_0 \prod_{i=0}^{\tau-1} \gamma_i + \eta \sum_{i=0}^{\tau-1} v \cdot \nabla H(w_i, x_i) \prod_{j=i+1}^{\tau-1} \gamma_j \\ w_{\tau+1} = w_0 \prod_{i=0}^{\tau-1} \delta_i + \eta \sum_{i=0}^{\tau-1} \left(\vec{R}_i + \nabla H(w_i, x_i)\right) \prod_{j=i+1}^{\tau-1} \delta_j \end{cases} \tag{56}$$

We will now prove the statement by induction: so assume that all $w_i$ and $\alpha_i$ satisfy the statement up to $i = t$ and we prove it for $t + 1$. So we will use that $\gamma_i, \delta_i \leq 1 - \eta k_0$ and $\alpha_i \leq \alpha_0 \to_d 0$. Now we estimate the noise terms that luckily in eq. (56) appear as geometrically dampened martingale: h

$$\vec{M}_\tau = \eta \sum_{i=0}^{\tau-1} \nabla H(w_i, x_i) \prod_{j=i+1}^{\tau-1} \gamma_\tau.$$

Using Doob inequality, together with assumption we get:

$$\mathbb{P}\left(\sup_{u \in \mathbb{S}^{d-1}} \sup_{\tau \leq n} u \cdot \vec{M}_\tau \geq r\right) \leq \frac{\mathrm{Var}(u \cdot M_n)}{r^2} = \frac{\eta^2 C_1 ||w_0||^2}{r^2} \sum_{i=0}^{n} (1 - \eta k_0)^{2i} = \frac{\eta C_1 ||w_0||^2}{r^2} \frac{1 - (1 - \eta k_0)^{2n}}{2k_0 - \eta k_0^2} \tag{57}$$

where we used the fact that $\mathrm{Var}(\nabla H^2(x, w) \leq C ||w||^2$ that can be quickly checked to be true. So pick a sequence of $r_d \to 0$ as $d \to \infty$ and condition to the complementary event in *eq.* (57). Then, we can take $d$ large enough so that $\alpha_0 \leq \frac{k_0}{2L}$, so that $\gamma_\tau \leq 1 - \eta \frac{k_0}{2} = \bar{\gamma}$, plugging into the first equation in eq. (56) we find :

$$\alpha_{\tau+1} \leq \alpha_0 \bar{\gamma}^{\tau+1} + r_d$$

So by taking $d$ large so that $r_d$ becomes small enough we have verified the requirement on $\alpha_{\tau+1}$. We can turn now to verify the inductive statement for $w_{\tau+1}$. Applying the inductive hypotheses on eq. (56) we get:

$$||w_{\tau+1}|| \leq ||w_0|| \bar{\delta}^{\tau+1} + \eta \sum_{i=0}^{\tau-1} L\alpha_i \bar{\delta}^{\tau-i} + \left\|\eta \sum_{i=0}^{\tau-1} \nabla H(w_i, x_i) \bar{\delta}^{\tau-i}\right\| \tag{58}$$

we can estimate the middle term by applying the inequality for $\alpha_i$ and getting (taking again $d$ large so that $r$ is small enough):

$$\eta \sum_{i=0} L\alpha_i \bar{\delta}^{\tau-i} \leq \underbrace{\eta L\alpha_0 (\min \bar{\gamma}, \bar{\delta})^\tau}_{\to 0} + L\frac{r}{k_0}$$

Finally the last term in eq. (58) can be estimate by Chebyshev inequality since

$$\mathrm{Var}\left(\eta \sum_{i=0}^{\tau-1} \nabla H(w_i, x_i) \bar{\delta}^{\tau-i}\right) \leq C \frac{\eta^2}{1 - \bar{\delta}^2} \leq C \frac{||w_0||^2 \eta}{k_0}$$

which tends to 0 as $d \to \infty$. Note that we used the inductive hypothesis to estimate $\text{Var}(\nabla H(x,w)) \le C||w_0||$. So we can establish that:

$$\mathbb{P}\left(\left\|\eta \sum_{i=0}^{\tau-1} \nabla H(w_i, x_i)\bar{\delta}^{\tau-i}\right\| \ge r_d\right) \le C\frac{||w_0||^2\eta}{r_d^2 k_0} \tag{59}$$

This concludes the verification of the inductive step. Note that up to now we were conditioning to be in the complementary of eq. (57) and of eq. (59), but note that if we take $r_d$ to go to 0 slowly enough so that $\frac{\eta}{r^2} \to 0$, then the probability of the event in eq. (57) and eq. (59) goes to 0 and the proof is concluded.

$\square$

### B.8. Trainable intensity of the skip connection

Let us see what changes if we add a weight $b$ that multiplies the intensity of the skip connection and train it.

$$-\nabla_w \mathcal{L}_t = \tag{60}$$

$$-= \mathbb{E}_{x,z}\left[\sigma'(x_w)\sigma(x_w)||w||^2 x + \sigma^2(x_w)w + b\sigma(x_w)x + b\sigma'(x_w)x_w x - \frac{1}{\sqrt{\Delta_t}}\left(\sigma(x_w)z + \sigma'(x_w)z_w x\right)\right] \tag{61}$$

$$= -\mathbb{E}_x\left[x\left(\underbrace{\sigma''(x_w)||w||^2 - \sigma'(x_w)\sigma(x_w)||w||^2 - b\sigma(x_w) - b\sigma'(x_w)x_w}_{F_\sigma}\right) + w\left(\underbrace{2\sigma'(x_w) - \sigma^2(x_w)}_{G_\sigma}\right)\right] \tag{62}$$

We also have the derivative with respect to $b$:

$$-\nabla_b \mathcal{L}_t = -\mathbb{E}_{x,z}\left[\sigma(x_w)x_w + b||x||^2 - \frac{1}{\sqrt{\Delta_t}}\left(x^\top z\right)\right] \tag{63}$$

$$= -\mathbb{E}_{x,z}\left[\sigma(x_w)x_w + b||x||^2 - 1\right] \tag{64}$$

Since $\text{Var}(x) = 1$, $(w, b) = (0, 1)$ is a critical point, and around it the dynamics will be similar to the previously examined case: from eq. (64) we can see that the dynamics attract towards $b = 1$.

