# OpenReview forum: "A theory of learning data statistics in diffusion models, from easy to hard"
_ICML.cc/2026/Conference — ICML 2026 regular_

### Official Review · Reviewer_hfe8 · 2026-03-03

**Soundness:** 3
**Presentation:** 3
**Significance:** 4
**Originality:** 3
**Overall Recommendation:** 5
**Confidence:** 4

**Summary:**

This paper analyzes the complexity of learning correlations of increasing order by training denoisers using SGD. The authors begin with an experiment showing that a neural denoiser's behavior on real-world images initially matches its behavior on a Gaussian approximation to the data distribution early in training, but diverges as training progresses. Motivated by these findings, the authors study the dynamics of projected SGD on a simple non-linear denoising architecture trained on data from a mixed cumulant model (MCM) -- a generalization of Gaussian data that can model higher-order correlations "hidden" in otherwise Gaussian data. Their key theoretical finding is that these dynamics are governed by a "diffusion information exponent" $k^*$, which depends on both the architecture and the data distribution. They also theoretically study the impact of the spherical constraint in projected SGD and empirically study the role of depth on the learning dynamics.

**Compliance With Llm Reviewing Policy:**

Affirmed.

**Final Justification:**

I maintain my positive assessment of this paper and continue to recommend acceptance.

**Key Questions For Authors:**

1. What accounts for the empirical threshold of $10^3$ samples before the denoiser's performance on the test set diverges from its performance on the clone dataset? Is this dependent on the structure of the dataset, its size, its dimensionality, etc?

2. I believe that the optimal denoiser for Gaussian data with mean $\mu$ and covariance $\Sigma$ is a Wiener filter -- i.e. a linear map. In particular, the optimal denoiser for the clone datasets in Section 2 has this linear structure. Can we interpret this section's findings as demonstrating that neural denoisers behave essentially like optimal linear denoisers early in training? If so, this seems closely connected to the findings in Li et al. (2024), which the authors cite in the introduction. I wonder if

3. How should we understand the MCM's spike directions $u,v$ in the context of real-world data? Can we understand these spikes are modeling phenomena that we observe in such data? For example, it seems to me that in the context of image data, we can perhaps understand $v$ as a "global filter" that picks out interesting features of images. In the MCM model, the distribution of $\langle x,v \rangle$ is non-Gaussian (e.g. potentially multimodal), which can perhaps model data where only a few outliers activate the filter.

4. Suppose we only have access to a fixed number of samples and cannot run online SGD. Do the results in this paper have any straightforward implications on the transition from generalization to memorization in diffusion models?

**Limitations:**

Yes

**Strengths And Weaknesses:**

This is a valuable contribution to the theory of diffusion models that provides a theoretical complement to recent empirical findings by Favero et al. (2025) and Bonnaire et al. (2025) concerning the generalization of diffusion models across gradient descent iterations. The mixed cumulant model is a clever relaxation of the Gaussian-data assumption that is frequently made in diffusion theory which enables the authors to study the sample complexity of learning higher-order structure (i.e. beyond the first two moments) using diffusion models. I expect that similar techniques could be used to theoretically study generalization in diffusion models, which is a central problem in the theory of diffusion models.

The paper is well-written, and the main ideas are generally easy to follow. That said, authors could devote some space to visualizing and motivating the mixed cumulant model, which appears somewhat unnatural at first glance. One of my questions below seeks to clarify my understanding of this model.

To the best of my knowledge, this paper is technically sound -- though I have not read the proofs in great detail. The experiment in Section 2, whose results are presented in Figure 1, does a good job of motivating the analysis in the remainder of the paper.

---

> ### Author Rebuttal · Authors · 2026-03-30
>
> We are grateful to the reviewer for the positive assessment, for recognizing the value of the mixed cumulant model as a tool for studying learning in diffusion models, and for the insightful questions that point toward natural extensions of our work.
>
> **Empirical threshold for divergence from Gaussian clones (Q1).** The threshold at which the denoiser's performance on real data separates from the Gaussian clone depends on all three factors the reviewer mentions. Our theory predicts that the threshold scales with dimension $d$ according to the diffusion information exponent $k^*$, and is modulated by the strength of the relevant cumulant coefficients $c_i^L$ (which encode the data structure) and the learning rate. In practice, the separation also depends on how much the data's higher-order statistics "overlap" with its covariance structure: as we discuss in Prop. 4.6 part 2, correlated latent variables can dramatically accelerate the learning of higher-order features.
>
> **Connection to optimal linear denoisers and Li et al. (2024) (Q2).** The reviewer's interpretation is exactly right. Early in training, before the network has learned to exploit higher-order statistics, it effectively acts as a linear denoiser. Li et al. (2024) showed that diffusion models in the generalization regime exhibit an inductive bias towards Gaussian structure: their trained nonlinear denoisers are well approximated by a linear model that corresponds to the Wiener filter, i.e. the optimal denoiser for the Gaussian distribution matching the data's mean and covariance. This is precisely what we observe in the initial phase of training in our experiments, where the loss on real data and on the Gaussian clone coincide. Our work is naturally complementary: while Li et al. characterize and document this Gaussian/linear phase, we provide a theory for *when and how* the denoiser transitions beyond it.
>
> **Interpreting MCM spikes for real data (Q3).** The reviewer's intuition about "global filters" is  a good perspective. A natural way to think about the spike directions $u$ and $v$ in image data is through the lens of Fourier modes or Gabor-like filters. For instance, the covariance spike $u$ can be understood as a dominant Fourier mode (e.g. a low-frequency component capturing overall brightness or large-scale structure), which contributes strongly to the covariance but has roughly Gaussian projections. The cumulant spike $v$ could correspond to a more localized, oriented Gabor filter that picks up edge-like features: projections along such directions are typically sparse and heavy-tailed (i.e. non-Gaussian), since most image patches are smooth but a few contain sharp edges that strongly activate the filter. The MCM captures, in a minimal way, the coexistence of these two types of structure. We agree that adding a paragraph motivating the MCM through such examples would improve the paper and will do so in the revision.
>
> **Implications for memorization with fixed samples (Q4).** Our theory uses simple architectures that cannot memorize, so the generalization-memorization tradeoff does not fully manifest in our setting; studying it properly would require richer, overparameterized models. That said, the fact that different statistical features have different sample complexities (governed by $k^*$) is suggestive: it is tempting to speculate that the transition from generalization to memorization could occur feature-by-feature, depending on its statistical properties. We find this a compelling direction for future work.
>
>
> We thank the reviewer again for the supportive assessment and the stimulating questions, which have helped us identify concrete improvements for the manuscript.

---

> > ### Author Rebuttal · Reviewer_hfe8 · 2026-04-02
> >
> > Thank you for these helpful responses. I maintain my accept rating for this manuscript.

---

### Official Review · Reviewer_VrDs · 2026-03-04

**Soundness:** 3
**Presentation:** 3
**Significance:** 2
**Originality:** 2
**Overall Recommendation:** 4
**Confidence:** 4

**Summary:**

The paper provides evidence that diffusion models learn data structure from easy to hard: during training, they first fit pairwise statistics  (mean and covariance) and only later exploit higher-order correlations. This behavior is captured by a toy denoiser trained with SGD on mixed-cumulant data, where the difficulty of learning is controlled by the diffusion information exponent. The paper proves a sample-complexity gap: learning pairwise statistics can require a number of samples that scales linearly with the dimension, whereas learning higher-order structure can require at least a cubic number of samples. Furthermore, when pairwise and higher-order statistics share correlated latent structure, higher-order features become easier to learn.

**Compliance With Llm Reviewing Policy:**

Affirmed.

**Final Justification:**

I find the paper sound and worthwhile within its scope, with significance somewhat limited by the gap between the theoretical setting and practical training setups, and I am therefore increasing my recommendation from 3 to 4, as my main concerns were addressed by the authors.

**Key Questions For Authors:**

- In Figure 1, are we observing a data-statistics effect or an optimization effect?
- Could the authors validate the linear vs cubic sample-complexity claim by training with different dataset sizes $n$ and checking when the model starts outperforming the mean+covariance clone?
- In the experiments, the authors compare loss on real data to loss on Gaussian clones matched in mean and covariance. Is the gap to the real-data loss a reliable proxy for learning higher-order structure? Why not construct clones that also match higher-order moments? Is the main obstacle computational cost, or is it conceptually difficult for images?
- How sensitive is this phenomenon to the noise schedule and the choice of loss weighting over $t$?
- The paper claims that correlated latent factors can reduce the sample complexity of learning higher-order structure. What might such correlated latent structure look like in real datasets?

**Limitations:**

Yes

**Strengths And Weaknesses:**

**Strengths**

- The easy to hard learning phenomenon is easy to understand.
- The paper provides an explanation for why learning higher-order structure has high sample complexity, and it also identifies conditions under which this sample complexity can be reduced.
- The claims are supported by experiments using U-Net architecture on image datasets.

**Weaknesses**

- The theory relies on a simplified denoiser and a restrictive data model, so it is unclear how the derived sample-complexity bounds extend to the general case.
- Figure 1 and Figure 6 look like an optimization-time effect: early iterations fit mean/covariance, and only later iterations exploit higher-order structure. But it seems not directly support the linear vs cubic sample-complexity result.
- The experiments do not change the dataset size or the dimension $d$, so they do not directly show how the required data grows with $d$.

---

> ### Author Rebuttal · Authors · 2026-03-30
>
> We thank the reviewer for the thoughtful assessment, for recognizing the clarity of the easy-to-hard phenomenon, and for the constructive questions.
>
> **How do our results extend to the general case? (Weakness 1)** Our theory is proven for online projected SGD on a single-neuron denoiser trained on synthetic data, while our experiments (Figs. 1, 6) use U-Nets trained with Adam on natural images. The purpose of this two-level approach is intentional: the theory identifies the core mechanism in a controlled setting, and the experiments demonstrate that the same qualitative phenomenon persists in realistic architectures, optimizers, and data. We do not claim a quantitative mapping between the two.
>
> **Data-statistics effect vs. optimization effect (Q1, Weakness 2).** This is not an either/or distinction. Our theory explains how different data statistics are picked up at different stages of optimization: $k^*$ governs which features become learnable at which sample complexity, and in online SGD each step uses a fresh sample, so sample complexity and optimization time are directly linked. The sequential learning in Figs. 1 and 6 is an optimization effect *driven by* the data statistics.
>
> **Validating scaling by varying dataset size/dimension (Q2, Weakness 3).** We stress that we do *not* expect cubic scaling on natural images, because this applies only when pairwise and higher-order statistics are carried by *independent* latent directions. With correlated latents (Prop. 4.6 part 2), sample complexity drops to quasi-linear $\Theta(d \ \mathrm{polylog}(d))$. Real data almost certainly falls in this regime (see Q5 below).
>
> To provide additional empirical support of our theoretical predictions, we trained two-layer ReLU autoencoders with SGD on MCM data at varying $d$ (Rebuttal Figure 1: https://figshare.com/s/634fe69994c9bfe317ab?file=63284353). Panel a) shows independent latents: overlap curves collapse when steps are rescaled by $d^{-3}$, confirming cubic complexity. Panel b) shows correlated latents: curves collapse with $d^{-1}$ rescaling, confirming linear complexity. These use overparameterized networks, demonstrating that Prop. 4.6 extends beyond the single-neuron model.
>
> **Higher-order clones (Q3).** Matching higher-order cumulants is not feasible in high dimension. The fourth cumulant tensor has $d^4$ entries, making estimation and constrained sampling prohibitive for $d \sim 10^3$--$10^4$. Even restricting to a few directions requires choosing *which* directions a priori, defeating the purpose. The Gaussian clone is clean because it captures *all* pairwise statistics and *no* higher-order ones. We will add a remark explaining this.
>
> **Sensitivity to noise schedule and loss weighting (Q4).** Our theory holds at each fixed $t$. Panels b)--c) of Fig. 1 and f)--g) of Fig. 6 already show sequential learning at individual noise levels, confirming it is not an artifact of averaging over $t$. The qualitative picture holds as long as the Hermite coefficients $c_i^L$ remain nonzero (e.g. $c_4^L = e^{-4t} - 3e^{-2t}$, Section B.4.2), which excludes only extreme noise levels that are uninteresting in practice.
>
> **Correlated latent structure in real datasets (Q5).** Consider face images. The shape of a human eye is non-Gaussian (sharp edges, specific intensity profile), captured only by higher-order statistics. But bilateral symmetry means two nearly identical eyes also create long-range pairwise correlations. The same latent factor thus generates both covariance and higher-order structure, exactly the correlated regime of Prop. 4.6 part 2. Of course, real data has far richer structure than any single factor can capture; this is an interpretable illustration of why the assumption of *completely independent* covariance and cumulant directions is likely too simplistic for natural data. Even partial overlap suffices to enter the correlated regime with quasi-linear sample complexity.
>
> We believe these clarifications address the reviewer's concerns and kindly ask the reviewer to reconsider the score. If not, we are happy to provide further clarifications during the discussion period.

---

> > ### Author Rebuttal · Reviewer_VrDs · 2026-04-02
> >
> > Thank you for the rebuttal and for addressing several of my questions. I appreciate the additional clarifications and analyses.
> >
> > In the figure provided in the rebuttal, what is kept fixed when $d$ changes, especially the learning rate and initialization? More generally, since the theory is about sample complexity, why not vary the number of training samples $n$ directly (that is, the dataset size / sample budget), instead of only varying $d$ and rescaling training steps? This seems like a more direct test of the sample-complexity claim.
> >
> > I understand your point that, in online SGD, data statistics and optimization are linked. Still, what kind of evidence would help show that the observed sequential learning is mainly driven by the statistics of the data, rather than mostly by the optimizer or training setup?
> > Could you also clarify more explicitly the difference between the training setting in the theory and in the main experiments? The theory studies online SGD with a fresh sample at each step, while the experiments seem to use standard repeated-pass minibatch training on a fixed dataset. How should the sample-complexity interpretation be understood across these two settings?

---

> > > ### Author Response · Authors · 2026-04-06
> > >
> > > We thank the reviewer for continuing to engage with our work.
> > >
> > > **Details of the rebuttal figure.** The learning rate is fixed at $10^{-1}$ across all $d$, and each curve averages over 5 independent initializations. These simulations use online SGD, so the number of steps equals the number of samples seen. Varying steps *is* varying the sample budget, and the collapse under $d^{-3}$ (panel a) and $d^{-1}$ (panel b) rescaling directly confirms the predictions of Prop. 4.6.
> > >
> > > We present **two lines of evidence to establish sequential learning is data driven**.
> > > - The first is **experimental**: in our experiment from Fig 1, we train a diffusion model on real images and evaluate it on a test set with CelebA test images, and on a test set with samples from a Gaussian distribution that has the same mean and covariance as the real images (the "clone"). The optimizer, architecture, and hyperparameters are identical across evaluations; the only difference is the structure of the evaluation data, which reveals the sequential learning effect: early on during training, the denoiser achieves the same test loss on real images and on Gaussian samples, which means that the denoiser only relies on pair-wise correlations; only after about $10^3$ SGD steps, the test losses diverge, meaning the network starts to exploit the higher-order correlations between pixels.
> > > - The second line is **theoretical**: our analysis gives a precise mechanism for why the sequential learning occurs: the population loss decomposes into contributions from different orders of input statistics (via the Hermite expansion), and the low-order terms (mean and covariance) coincide between real data and Gaussian clones by construction, while higher-order terms are present only in the real data. The timescale at which these terms become learnable is governed by the diffusion information exponent $k^*$ that we introduce. Our theoretical results precisely predict what we observe experimentally: the loss on real data first matches the clone (while only pairwise statistics are being exploited), then separates (when higher-order structure kicks in). It is very difficult to see how pure optimization effects could produce this pattern, and in particular how they could produce the different behavior we observe for clones matching only the mean vs. clones matching mean and covariance.
> > >
> > > **Online SGD vs. repeated-pass minibatch training.** Our theoretical results are stated for online SGD, where sample complexity and optimization time coincide, which makes the analysis tractable and is a standard setting in neural network theory, see refs [1-4] for some recent examples. Our primary interest is the *learning dynamics*, i.e. the ordering in which features are learned, rather than exact sample complexity thresholds. In our experiments, we use standard practical training setups (Adam, standard learning rates, standard initialization) precisely because we want to understand the learning dynamics as they occur in practice. For CelebA ($\sim 10^5$ samples, batch size $10^2$), one epoch is $\sim 10^3$ steps, and the real-data and clone losses already diverge around this point, meaning the model begins exploiting higher-order statistics before it has even completed a single pass over the data. This makes the effective setting close to online learning. A full theoretical treatment of the multi-pass regime is an interesting open direction, but the ordering of feature learning predicted by our theory is clearly reflected in the experiments.
> > >
> > > [1] Ben Arous et al. NeurIPS 2022 (best paper award)
> > >
> > > [2] Damian et al. NeurIPS 2023
> > >
> > > [3] Bardone & Goldt ICML 2024
> > >
> > > [4] Lee et al. NeurIPS 2024

---

### Official Review · Reviewer_cjXh · 2026-03-09

**Soundness:** 2
**Presentation:** 3
**Significance:** 2
**Originality:** 2
**Overall Recommendation:** 4
**Confidence:** 3

**Summary:**

This paper studies learning dynamics in denoising diffusion models in a mixed cumulant model setting, proposing a diffusion information exponent similar to multi/single-index model literature.

**Compliance With Llm Reviewing Policy:**

Affirmed.

**Final Justification:**

The authors have sufficiently addressed my concerns.

**Key Questions For Authors:**

See weakness above.

**Limitations:**

No. The paper includes an impact statement that essentially says there are potential societal consequences but none worth highlighting. I suggest the authors add a short paragraph discussing the limitations of analysis under MCMs and whether this data distribution is representative.

**Strengths And Weaknesses:**

Strength:
1. The paper is clearly written and generally readable. The model definition and target (score function) is clearly presented, where the target resembles a single-index target.
2. The paper gives both sharp upper/lower bound (postive& negative results) on the dynamics, showing exactly when the diffuion model learns the target.

Weakness:
I find some mathematical derivation confusing:
1. Equation (8) is dimensionally wrong. In eq. 8 & 9, Is $F_σ(x · w)$ a scalar or a vector?
2. In Proposition 4.6 (Line 309), it only rules out recovery up to a linear regime $\min(d/\eta_{d}^2,d)$. Why does pSGD need $d^{3}$ samples?
3. The proof in Appendix B.6 is hard to follow. This subsection derives the dynamics $-\nabla L_t(w)$, and then claims that the dynamics are governed by $\Lambda$. The extra factor in $\Lambda$ ($1+c^L_2$) appears from nowhere.
4. Assumption 4.5 contains a false implication. The boundedness of $\sigma$'s derivatives do not by themselves imply this F is globally Lipschitz, because $F$ contains the term $-\sigma ' (z)z$. Perhaps the intended activations (e.g. tanh/sigmoid) make it true, but the stated implication is false in general. You need to ensure $\sigma$'s derivatives not flucturate when $z\rightarrow \infty$.

A more high-level weakness is:
The paper doesn't argue that mixed cumulant model is a representative and novel model to analyze the dynamics of diffusion models. In this setting, the score function is pretty similar to a single-index target (with a slightly non-Gaussian input), which makes the paper less relavent to the diffusion model itself, more like an incremental work of Bardone and Goldt (2024).

---

> ### Author Rebuttal · Authors · 2026-03-30
>
> We sincerely thank the reviewer for their careful reading, for acknowledging the clarity of the writing and the sharpness of our bounds, and especially for the detailed scrutiny of the mathematical derivations.
>
> **Typo in Eq. (8).** We thank the reviewer for spotting this. Indeed, $F_\sigma$ is a scalar, and the vector $x$ was mistakenly omitted. The corrected equation reads:
> $$\nabla_{\text{sph}} \mathscr{L}_t(w,x) = (\mathrm{Id} _ d - ww^\top) x F _ \sigma(x \cdot w).$$
> This is fixed in the revised manuscript.
>
> **Missing exponent in Proposition 4.6.** Another crucial typo, thank you. The negative result in Prop. 4.6 part 1 should read $\min(d/\eta_d^2, d^3)$, not $\min(d/\eta_d^2, d)$. The exponent 3 was dropped during typesetting. This is consistent with the cubic sample complexity discussed in Section 4.2 and with Prop. 4.3 at $k^*=4$. Corrected in the revision.
>
> **Proof in Appendix B.6.** We agree this step deserved more explanation. The factor $(1+c_2^L)$ arises from two distinct terms in Eq. (B.19): $c_1^F$ comes from the leading Hermite coefficient of $\tilde{F}_\sigma$, while $c_2^L c_1^F$ comes from the rightmost cross-term (nonzero only when $k^* \le 2$). We will add a clarifying sentence in the revised appendix.
>
> **Assumption 4.5 and Lipschitz claim.** Good point. Boundedness of $\sigma$'s derivatives does not imply $F$ is globally Lipschitz because of the term $\sigma'(z)z$. However, our proofs only require $F \in L^2(\mathbb{R})$ (Gaussian-weighted), for which at most quadratic growth suffices. The Lipschitz claim is unnecessary and will be removed. We thank the reviewer for this precise observation.
>
> **Representativeness of the mixed cumulant model** While it is of course a highly stylised view of the true distribution of images, the mixed cumulant model (MCM) captures several salient features of images: it generates non-Gaussian inputs; both the covariance _and_ the higher-order cumulants are anisotropic, and they carry distinct statistical signals (the principal components of real images look typically like Fourier modes, while principal components of higher-order cumulants resemble localised Gabor filters). Many recent works on diffusion models have considered training data sampled from isotropic Gaussian $\mathcal{N} (0, id_d)$, for example the best-paper-award-winning work of Bonnaire et al. at NeurIPS '25, while even works that consider structured inputs have focused on Gaussian inputs only, which strongly simplifies the analysis, see e.g. Wang & Pehlevan, NeurIPS '25. We therefore believe that studying diffusion models on a non-Gaussian input model like the MCM is a crucial step forward in our understanding of these models.
>
> **Novelty relative to Bardone & Goldt (2024)** While we do use the mixed cumulant model of Bardone & Goldt (2024), we would like to respectfully push back against the notion that this limits the novelty of our contribution. First of all, sharing a data model is very common in the theory of machine learning: there are virtually hundreds of papers at NeurIPS / ICML / ICLR that consider supervised learning of random target functions over Gaussian inputs; sharing this input model does not automatically make these papers incremental with respect to each other. Specifically, Bardone & Goldt ('24) analysed supervised *classification* on the mixed cumulant model; here, we study *score denoising* in a *diffusion* setting. The task, the structure of the loss, and the resulting phenomenology are fundamentally different and yield qualitatively new phenomena with no counterpart in the supervised setting. For example,
> 1. The matched denoiser $\sigma^*_t$ (Section 4.4) provides the first example we are aware of where matching the ground-truth data model is *suboptimal* with respect to the choice of nonlinearity in the denoiser;
> 2. On a technical level, the MSE denoising loss introduces an additional contraction term ($\sigma^2$ inside $G_\sigma$) and the Stein-lemma contribution ($2\sigma'$ in $G_\sigma$), both absent in supervised correlation losses. These give rise to the trapping at $w=0$ for unconstrained SGD (Prop. 4.7), a novel finding.
>
> **Limitations discussion.** We discuss limitations in Section 5, noting the restriction to a single hidden unit and single non-Gaussian direction, and the gap between our theoretical sample complexity and the rapid separation observed experimentally. We will expand this discussion into a dedicated limitiations section with clear heading, while also discussing the representativeness of the MCM.
>
> We believe the concerns of the reviewer are fully addressable through these corrections and clarifications: typos are fixed, the Lipschitz claim in assumption 4.5 addressed, and the novelty beyond prior work clarified. We kindly ask the reviewer to consider whether these responses resolve the identified weaknesses and whether an upward revision of the score may be appropriate.

---

> > ### Author Rebuttal · Reviewer_cjXh · 2026-04-04
> >
> > The authors have sufficiently addressed my concerns. I will raise my score to 4.

---

### Official Review · Reviewer_kDKL · 2026-03-12

**Soundness:** 2
**Presentation:** 2
**Significance:** 3
**Originality:** 3
**Overall Recommendation:** 4
**Confidence:** 4

**Summary:**

This article studies an simplicity bias in the learning dynamics of diffusion models, where pairwise input statistics are learnt before specializing to higher-order correlations. This behavior is studied in a simple scenario where denoisers are trained on a mixed cumulant model. Theoretical results are given in high-dimensional regime. It is found that depending on the number of training samples used in the dynamics, the learnt score function capture differently the input statistics. These results provide a theorical explanation of the practical simplicity bias in diffusion models.

**Compliance With Llm Reviewing Policy:**

Affirmed.

**Final Justification:**

na

**Key Questions For Authors:**

Technical clarify:
- Do you assume the w in eq 6 to be on the sphere (S^{d-1})? It is not so clear.
- In Proposition 4.3, what is the w(d), w(dlog^2d)? I am not sure how to interpret the value w(hat(n)(d,k*)), since hat(n) relies on the value of w which is learnt, and not known a-prior…
- In Proposition 4.4, what is the w(t) in eq 14? Should it be w_\tau?
- It is hard to interpret the second point in Proposition 4.6, do you need to assume \eta > 0 and close to be 1?
- Do you assume in section 4.4, u=0? If seems that you could w=v, so there is no impact from the u on the solution w. I am not sure if you assume u=0 also in section 3.3. This could be clarified.


The notation of SDE in eq 1 is not so common in the literature of stochastic process. I would use dx(t) for this purpose.

Typo: line 209 Thrice -> twice?

**Strengths And Weaknesses:**

The idea to study the mixed cumulant model in the context of learning diffusion models is quite original. By analyzing the situation as the input data dimension goes to infinity, the learning dynamics of projected SGD (vs. SGD) becomes explicit, providing new insights on the learnt score function at a given diffusion time.

The writing could still be improved. Some of the texts in the introduction are not finished (e.g. line 100 with …). My main concern is about some technical points, which should be clarified (see the questions below).

---

> ### Author Rebuttal · Authors · 2026-03-30
>
> We thank the reviewer for their careful reading and for recognizing the originality of studying the mixed cumulant model in the diffusion context. We address all questions below.
>
> **Q1: Is w in eq. (6) on the sphere?**
>
> Yes, in Sections 3, 4.1, and 4.2, $w$ is maintained on $\mathbb S ^{d-1}$ via projected SGD (eq. 7), which normalizes after each update, with initialization $w_0 \sim \text{Unif}(\mathbb S^{d-1})$. However, in Section 4.3 we explicitly remove the spherical constraint to study standard (unconstrained) SGD (eq. 17), which is a key part of our analysis showing that the spherical projection is crucial for efficient learning. We will make this distinction clearer around eq. (6).
>
> **Q2: What is $ω(d)$, $ω(d log^2d)$ in Proposition 4.3?**
>
> We apologize for the confusion. Here ω(·) denotes the standard little-omega asymptotic notation (i.e., ω(f(d)) means *"any function growing strictly faster than f(d)"*). In particular, $n̂(d, k^\star)$ is a *sample complexity threshold* that depends only on the dimension $d$ and the information exponent $k^\star$, and it does not depend on the learned weight $w$. For example, $n̂(d,1) = ω(d)$ means the number of samples must grow faster than linear in $d$. We will clarify the notation.
>
> **Q3: $w(t)$ in eq. (14) of Proposition 4.4**
>
> Yes, this should read $w_τ$ to be consistent with the rest of the paper. Thank you for catching this typo; we will fix it.
>
> **Q4: Interpretation of Proposition 4.6, point 2**
>
> $η$ is not assumed to be close to 1. The key point is that *any* positive constant $η$ independent of $d$ suffices. This is what we call "weak recovery": it means that the overlap $w_τ · u$ stays bounded away from zero as $d → ∞$, so the high-dimensional optimization problem effectively reduces to a low-dimensional dynamical system. Once weak recovery is established, standard finite-dimensional arguments can be used to study the resulting dynamical system. We also note that $η$ here is not the learning rate (which is $η_d$); we will rename one of the two to avoid confusion.
>
> **Q5: Is $u = 0$ assumed in Section 4.4?**
>
> Section 4.4 discusses the optimal denoiser $σ^*$ for the single-spike (cumulant-only) setting, i.e., $β_u = 0$. The score in eq. (44) (derived in Appendix B.3) corresponds to this case. More broadly, $β_u = 0$ is assumed for the entire analysis of SGD without the spherical constraint starting at Section 4.3 (as written at lines 333-334), and Section 4.4 inherits this assumption. We will  emphasized more this point at the beginning of section 4.4.
>
> **Q6: SDE notation in eq. (1)**
>
> We agree that $dx(t) = −x dt + dW_t$ is more standard and will adopt it.
>
> **Q7: "Thrice" at line 299**
>
> We believe the reviewer refers to "thrice differentiable" in Assumption 4.5 (line 299) (since it is the only time the word appears in the document). This is intentional: the third derivative of $σ$ is needed because $F(z)$ involves $\sigma''$, whose growth estimates require control of $\sigma'''$. We will add a brief justification in the text to make this clearer.
>
> **Regarding writing quality:**  The "..." on line 100 is a stylistic choice to introduce the numbered list of contributions that follows. We understand the confusion this may have created and will remove the "...".
>
> We hope these clarifications address the reviewer's concerns. We believe the core technical issues raised are notational and presentational rather than substantive, and we will incorporate all suggested improvements in the revision. We kindly ask the reviewer to consider revising their score in light of our responses.

---

> > ### Author Rebuttal · Reviewer_kDKL · 2026-04-02
> >
> > Thanks for the clarification. I'd raise my score to accept the article.

---

### Decision · Program_Chairs · 2026-04-30

**Decision:**

Accept (regular)

**Comment:**

This paper demonstrates that diffusion models exhibit a simplicity bias, learning pairwise statistics before higher-order correlations, and provides a rigorous theory based on the "diffusion information exponent" governing this transition. The authors prove a sample-complexity gap — linear for pairwise statistics, cubic for higher-order cumulants — and show this gap collapses to quasi-linear when latent structures are correlated. All reviewers agreed that the paper makes a clear theoretical contribution to the diffusion models. I thus recommend acceptance.